# Harnessing Mixed Offline Reinforcement Learning Datasets via Trajectory Weighting

**Zhang-Wei Hong & Pulkit Agrawal**
Massachusetts Institute of Technology
USA
{zwhong,pulkitag}@mit.edu

**Rémi Tachet des Combes[*] & Romain Laroche[*]**

remi.tachet@gmail.com
romain.laroche@gmail.com

## Abstract

Most offline reinforcement learning (RL) algorithms return a target policy maximizing a trade-off between (1) the expected performance gain over the behavior policy that collected the dataset, and (2) the risk stemming from the out-of-distribution-ness of the induced state-action occupancy. It follows that the performance of the target policy is strongly related to the performance of the behavior policy and, thus, the trajectory return distribution of the dataset. We show that in mixed datasets consisting of mostly low-return trajectories and minor high-return trajectories, state-of-the-art offline RL algorithms are overly restrained by low-return trajectories and fail to exploit high-performing trajectories to the fullest. To overcome this issue, we show that, in deterministic MDPs with stochastic initial states, the dataset sampling can be re-weighted to induce an artificial dataset whose behavior policy has a higher return. This re-weighted sampling strategy may be combined with any offline RL algorithm. We further analyze that the opportunity for performance improvement over the behavior policy correlates with the positive-sided variance of the returns of the trajectories in the dataset. We empirically show that while CQL, IQL, and TD3+BC achieve only a part of this potential policy improvement, these same algorithms combined with our reweighted sampling strategy fully exploit the dataset. Furthermore, we empirically demonstrate that, despite its theoretical limitation, the approach may still be efficient in stochastic environments. The code is available at https://github.com/Improbable-AI/harness-offline-rl.

## 1 Introduction

Offline reinforcement learning (RL) currently receives great attention because it allows one to optimize RL policies from logged data without direct interaction with the environment. This makes the RL training process safer and cheaper since collecting interaction data is high-risk, expensive, and time-consuming in the real world (e.g., robotics, and health care). Unfortunately, several papers have shown that near optimality of the offline RL task is intractable sample-efficiency-wise (Xiao et al., 2022; Chen & Jiang, 2019; Foster et al., 2022).

In contrast to near optimality, policy improvement over the behavior policy is an objective that is approximately realizable since the behavior policy may efficiently be cloned with supervised learning (Urbancic, 1994; Torabi et al., 2018). Thus, most practical offline RL algorithms incorporate a component ensuring, either formally or intuitively, that the returned policy improves over the behavior policy: pessimistic algorithms make sure that a lower bound on the target policy (i.e., a policy learned by offline RL algorithms) value improves over the value of the behavior policy (Petrik et al., 2016; Kumar et al., 2020b; Buckman et al., 2020), conservative algorithms regularize their policy search with respect to the behavior policy (Thomas, 2015; Laroche et al., 2019; Fujimoto et al., 2019), and one-step algorithms prevent the target policy value from propagating through bootstrapping (Brandfonbrener et al., 2021). These algorithms use the behavior policy as a stepping stone. As a consequence, their performance guarantees highly depend on the performance of the behavior policy.

---

[*]Work done while at Microsoft Research Montreal.

Due to the dependency on behavior policy performance, these offline RL algorithms are susceptible to the return distribution of the trajectories in the dataset collected by a behavior policy. To illustrate this dependency, we will say that these algorithms are anchored to the behavior policy. Anchoring in a near-optimal dataset (i.e., expert) favors the performance of an algorithm, while anchoring in a low-performing dataset (e.g., novice) may hinder the target policy's performance. In realistic scenarios, offline RL datasets might consist mostly of low-performing trajectories with few minor high-performing trajectories collected by a mixture of behavior policies, since curating high-performing trajectories is costly. It is thus desirable to avoid anchoring on low-performing behavior policies and exploit high-performing ones in mixed datasets. However, we show that state-of-the-art offline RL algorithms fail to exploit high-performing trajectories to their fullest. We analyze that the potential for policy improvement over the behavior policy is correlated with the positive-sided variance (PSV) of the trajectory returns in the dataset and advance that when the return PSV is high, the algorithmic anchoring may be limiting the performance of the returned policy.

In order to provide a better algorithmic anchoring, we propose to alter the behavior policy without collecting additional data. We start by proving that re-weighting the dataset during the training of an offline RL algorithm is equivalent to performing this training with another behavior policy. Furthermore, under the assumption that the environment is deterministic, by giving larger weights to high-return trajectories, we can control the implicit behavior policy to be high performing and therefore grant a cold start performance boost to the offline RL algorithm. While determinism is a strong assumption that we prove to be necessary with a minimal failure example, we show that the guarantees still hold when the initial state is stochastic by re-weighting with, instead of the trajectory return, a trajectory return advantage: $G(\tau_i) - V^\mu(s_{i,0})$, where $G(\tau_i)$ is the return obtained for trajectory $i$, $V^\mu(s_{i,0})$ is the expected return of following the behavior policy $\mu$ from the initial state $s_{i,0}$. Furthermore, we empirically observe that our strategy allows performance gains over their uniform sampling counterparts even in stochastic environments. We also note that determinism is required by several state-of-the-art offline RL algorithms (Schmidhuber, 2019; Srivastava et al., 2019; Kumar et al., 2019b; Chen et al., 2021; Furuta et al., 2021; Brandfonbrener et al., 2022).

Under the guidance of theoretical analysis, our principal contribution is two simple weighted sampling strategies: **Return-weighting (RW)** and **Advantage-weighting (AW)**. RW and AW re-weight trajectories using the Boltzmann distribution of trajectory returns and advantages, respectively. Our weighted sampling strategies are agnostic to the underlying offline RL algorithms and thus can be a drop-in replacement in any off-the-shelf offline RL algorithms, essentially at no extra computational cost. We evaluate our sampling strategies on three state-of-the-art offline RL algorithms, CQL, IQL, and TD3+BC (Kumar et al., 2020b; Kostrikov et al., 2022; Fujimoto & Gu, 2021), as well as behavior cloning, over 62 datasets in D4RL benchmarks (Fu et al., 2020). The experimental results reported in statistically robust metrics (Agarwal et al., 2021) demonstrate that both our sampling strategies significantly boost the performance of all considered offline RL algorithms in challenging mixed datasets with sparse rewarding trajectories, and perform at least on par with them on regular datasets with evenly distributed return distributions.

## 2 PRELIMINARIES

We consider reinforcement learning (RL) problem in a Markov decision process (MDP) characterized by a tuple $(\mathcal{S}, \mathcal{A}, R, P, \rho_0)$, where $\mathcal{S}$ and $\mathcal{A}$ denote state and action spaces, respectively, $R : \mathcal{S} \times \mathcal{A} \to \mathbb{R}$ is a reward function, $P : \mathcal{S} \times \mathcal{A} \to \Delta_{\mathcal{S}}$ is a state transition dynamics, and $\rho_0 : \Delta_{\mathcal{S}}$ is an initial state distribution, where $\Delta_{\mathcal{X}}$ denotes a simplex over set $\mathcal{X}$. An MDP starts from an initial state $s_0 \sim \rho_0$. At each timestep $t$, an agent perceives the state $s_t$, takes an action $a_t \sim \pi(.|s_t)$ where $\pi : \mathcal{S} \to \Delta_{\mathcal{A}}$ is the agent's policy, receives a reward $r_t = R(s_t, a_t)$, and transitions to a next state $s_{t+1} \sim P(s_{t+1}|s_t, a_t)$. The performance of a policy $\pi$ is measured by the expected return $J(\pi)$ starting from initial states $s_0 \sim \rho_0$ shown as follows:

$$J(\pi) = \mathbb{E}\left[\sum_{t=0}^{\infty} R(s_t, a_t) \;\middle|\; s_0 \sim \rho_0, a_t \sim \pi(.|s_t), s_{t+1} \sim P(.|s_t, a_t)\right]. \tag{1}$$

Given a dataset $\mathcal{D}$ collected by a behavior policy $\mu : \mathcal{S} \to \Delta_{\mathcal{A}}$, offline RL algorithms aim to learn a target policy $\pi$ such that $J(\pi) \geq J(\mu)$ from a dataset $\mathcal{D}$ shown as follows:

$$\mathcal{D} = \left\{ (s_{i,0}, a_{i,0}, r_{i,0}, \cdots s_{i,T_i}) \,\Big|\, \begin{matrix} s_{i,0} \sim \rho_0, & a_{i,t} \sim \mu(.|s_{i,t}), \\ r_{i,t} = R(s_{i,t}, a_{i,t}), & s_{i,t+1} \sim P(.|s_{i,t}, a_{i,t}) \end{matrix} \right\}, \tag{2}$$

where $\tau_i = (s_{i,0}, a_{i,0}, r_{i,0}, \cdots s_{i,T_i+1})$ denotes trajectory $i$ in $\mathcal{D}$, $(i, t)$ denotes timestep $t$ in episode $i$, and $T_i$ denotes the length of $\tau_i$. Note that $\mu$ can be a mixture of multiple policies. For brevity, we omit the episode index $i$ in the subscript of state and actions, unless necessary. Generically, offline RL algorithms learn $\pi$ based on actor-critic methods that train a Q-value function $Q : \mathcal{S} \times \mathcal{A} \to \mathbb{R}$ and $\pi$ in parallel. The Q-value $Q(s, a)$ predicts the expected return of taking action $a$ at state $s$ and following $\pi$ later; $\pi$ maximizes the expected Q-value over $\mathcal{D}$. $Q$ and $\pi$ are trained through alternating between policy evaluation (Equation 3) and policy improvement (Equation 4) steps shown below:

$$Q \leftarrow \arg\min_Q \mathbb{E}\left[ \left( r_t + \gamma \mathbb{E}_{a' \sim \pi(.|s_{t+1})} [Q(s_{t+1}, a')] - Q(s_t, a_t) \right)^2 \,\Big|\, \text{Uni}(\mathcal{D}) \right] \tag{3}$$

$$\pi \leftarrow \arg\max_\pi \mathbb{E}\left[ Q(s_t, a) \,\Big|\, \text{Uni}(\mathcal{D}), a \sim \pi(.|s_t) \right], \tag{4}$$

where $\mathbb{E}[\,\cdot\,|\,\text{Uni}(\mathcal{D})]$ denotes an expectation over uniform sampling of transitions.

## 3 PROBLEM FORMULATION

Most offline RL algorithms are anchored to the behavior policy. This is beneficial when the dataset behavior policy is high-performing while detrimental when the behavior policy is low-performing. We consider mixed datasets consisting of mostly low-performing trajectories and a handful of high-performing trajectories. In such datasets, it is possible to exploit the rare high-performing trajectories, yet the anchoring restrains these algorithms from making sizable policy improvements over the behavior policy of the mixed dataset. We formally define the return positive-sided variance (RPSV) of a dataset in Section 3.1 and illustrate why the performance of offline RL algorithms could be limited on high-RPSV datasets in Section 3.2.

### 3.1 POSITIVE-SIDED VARIANCE

Formally, we are concerned with a dataset $\mathcal{D} := \{\tau_0, \tau_1, \cdots \tau_{N-1}\}$ potentially collected by various behavior policies $\{\mu_0, \mu_1, \cdots \mu_{N-1}\}$ and constituted of empirical returns $\{G(\tau_0), G(\tau_1), \cdots G(\tau_{N-1})\}$, where $\tau_i$ is generated by $\mu_i$, $N$ is the number of trajectories, $T_i$ denotes the length of $\tau_i$, and $G(\tau_i) = \sum_{t=0}^{T_i-1} r_{i,t}$. To study the distribution of return, we equip ourselves with a statistical quantity: the positive-sided variance (PSV) of a random variable $X$:

**Definition 1** (Positive-sided variance). *The positive-sided variance (PSV) of a random variable $X$ is the second-order moment of the positive component of $X - \mathbb{E}[X]$:*

$$\mathbb{V}_+[X] \doteq \mathbb{E}\left[ (X - \mathbb{E}[X])_+^2 \right] \quad \text{with} \quad x_+ = \max\{x, 0\}. \tag{5}$$

The return PSV (RPSV) of $\mathcal{D}$ aims at capturing the positive dispersion of the distribution of the trajectory returns. An interesting question to ask is: *what distribution leads to high RPSV?* We simplify sampling trajectories collected by a novice and an expert as sampling from a Bernoulli distribution $\mathcal{B}$, and suppose that the novice policy always yields a 0 return, while the expert always yields a 1 return. Figure 1a visualizes $\mathbb{V}_+[\mathcal{B}(p)] = p(1 - p)^2$, which is the Bernoulli distribution's PSV as a function of its parameter $p$, where $p$ is the probability of choosing an expert trajectory. We see that maximal PSV is achieved for $p = \frac{1}{3}$. Both $p = 0$ (pure novice) and $p = 1$ (pure expert) leads to a zero PSV. This observation indicates that mixed datasets tend to have higher RPSV than a dataset collected by a single policy. We present the return distribution of datasets at varying RPSV in Figure 1. Low-RPSV datasets have their highest returns that remain close to the mean return, which limits the opportunity for policy improvement. In contrast, the return distribution of high-RPSV datasets disperses away from the mean toward the positive end.

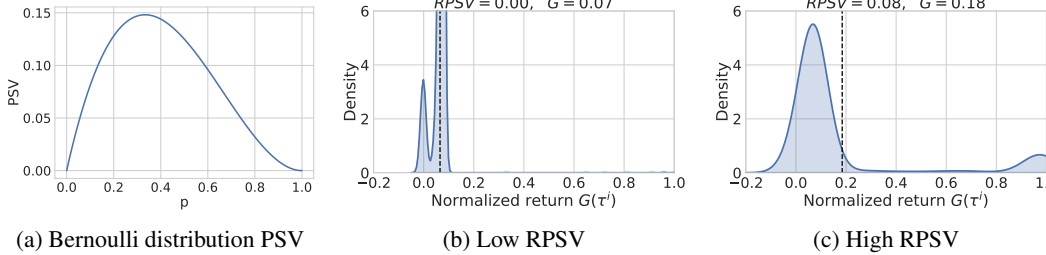

Figure 1: **(a)** Bernoulli distribution PSV: $\mathbb{V}_+[\mathcal{B}(p)] = p(1-p)^2$. **(b-c)** The return distribution of datasets with (b) low and (c) high return positive-sided variance (RPSV) (Section 3.1) , where RPSV measures the positive contributions in the variance of trajectory returns in a dataset and $\bar{G}$ denotes the average episodic returns (dashed line) of the dataset. Intuitively, a high RPSV implies some trajectories have far higher returns than the average.

### 3.2 OFFLINE RL FAILS TO UTILIZE DATA IN HIGH-RPSV DATASETS

High-RPSV datasets (Figure 1c) have a handful of high-return trajectories, yet the anchoring of offline RL algorithms on behavior policy inhibits offline RL from utilizing these high-return data to their fullest. Predominating low-return trajectories in a high-RPSV dataset restrain offline RL algorithms from learning a non-trivial policy close to the best trajectories in $\mathcal{D}$ due to these algorithms' pessimistic and/or conservative nature. High RPSV implies that the average episodic return is far from the best return in $\mathcal{D}$ (see Figure 1c). The average episodic return reflects the performance $J(\mu)$ (formally justified in Section 4.1) of the behavior policy $\mu$ that collected $\mathcal{D}$, where $\mu$ is mixture of $\{\mu_0, \mu_1, \cdots \mu_{N-1}\}$ (Section 3.1).

Pessimistic algorithms (Petrik et al., 2016; Kumar et al., 2020b; Buckman et al., 2020) strive to guarantee the algorithm returns a $\pi$ such that $J(\pi) \geq J(\mu)$, but this guarantee is loose when $J(\mu)$ is low. Conservative algorithms (Laroche et al., 2019; Fujimoto et al., 2019; Fujimoto & Gu, 2021; Kumar et al., 2019a) restrict $\pi$ to behave closely to $\mu$ to prevent exploiting poorly estimated Q-values on *out-of-distribution* state-action pairs in actor-critic updates (*i.e.*, $(s_{t+1}, a') \notin \mathcal{D}$ in Equation 3), hence restricting $J(\pi)$ from deviating too much from $J(\mu)$. Similarly, one-step algorithms (Brandfonbrener et al., 2021; Kostrikov et al., 2022) that perform only a single step of policy improvement return a target policy subject to constraints that enforces $\pi$ to be close to $\mu$ (Peters & Schaal, 2007; Peng et al., 2019). As a consequence, offline RL algorithms are restrained by $J(\mu)$ and fail to utilize high-return data far from $J(\mu)$ in high-RPSV datasets.

On the contrary, in low-RPSV datasets (Figure 1b), pessimistic, conservative, and one-step algorithms do not have this severe under-utilization issue since the return distribution concentrates around or below the average episodic return, and there are very few to no better trajectories to exploit. We will show, in Section 5.2, that no sampling strategy makes offline RL algorithms perform better in extremely low-RPSV datasets, while in high-RPSV datasets, our methods (Sections 4.2 and 4.3) outperform typical uniform sampling substantially.

## 4 METHOD

Section 3 explains why behavior policy anchoring prevents offline RL algorithms from exploiting high-RPSV datasets to their fullest. To overcome this issue, the question that needs to be answered is: *can we improve the performance of the behavior policy without collecting additional data?* To do so, we propose to implicitly alter it through a re-weighting of the transitions in the dataset. Indeed, we show that weighted sampling can emulate sampling transitions with a different behavior policy. We analyze the connection between weighted sampling and performance of the implicit behavior policy in Section 4.1, and then present two weighted sampling strategies in Sections 4.2 and 4.3.

### 4.1 ANALYSIS

We start by showing how re-weighting the transitions in a dataset emulates sampling transitions generated by an implicit mixture behavior policy different from the one that collected the dataset. It is implicit because the policy is defined by the weights of transitions in the dataset. As suggested

in Peng et al. (2019), sampling transitions from $\mathcal{D}$ defined in Section 3 is equivalent to sampling state-action pairs from a weighted joint state-action occupancy: $d_{\mathcal{W}}(s, a) = \sum_{i=0}^{N-1} w_i d_{\mu_i}(s) \mu_i(a|s)$, where $w_i$ is the weight of trajectory $i$ (each $\tau_i$ is collected by $\mu_i$), $\mathcal{W} \doteq \{w_0, \cdots w_{N-1}\}$, and $d_{\mu_i}(s)$ denotes the unnormalized state occupancy measure (Laroche et al., 2022) in the rollout of $\mu_i$. Tweaking weighting $\mathcal{W}$ effectively alters $d_{\mathcal{W}}$ and thus the transition distribution during sampling. As Peng et al. (2019) suggested, a weighting $\mathcal{W}$ also induces a weighted behavior policy: $\mu_{\mathcal{W}}(a|s) = \frac{d_{\mathcal{W}}(s,a)}{\sum_{i=0}^{N-1} w_i d_{\mu_i}(s)}$. Uniform sampling $w_i = \frac{1}{N}$, $\forall w_i \in \mathcal{W}$ is equivalent to sampling from the joint state-action occupancy of the original mixture behavior policy $\mu$ that collected $\mathcal{D}$. To obtain a well-defined sampling distribution over transitions, we need to convert these trajectory weights $w_i$ to transitions sample weights $w_{i,t}, \forall t \in [\![0, T_i - 1]\!]$:

$$w_{i,t} \doteq \frac{w_i}{\sum_{i=0}^{N-1} T_i w_i}, \qquad \sum_{i=0}^{N-1} \sum_{t=0}^{T_i-1} w_{i,t} = \sum_{i=0}^{N-1} T_i \frac{w_i}{\sum_{i=0}^{N-1} T_i w_i} = 1. \tag{6}$$

Thus, we formulate our goal as finding $\mathcal{W} \doteq \{w_i\}_{i \in [\![0, N-1]\!]} \in \Delta_N$ such that $J(\mu_{\mathcal{W}}) \geq J(\mu)$, where $\Delta_N$ denotes the simplex of size $N$. Naturally, we can write $J(\mu_{\mathcal{W}}) = \sum_{i=0}^{N-1} w_i J(\mu_i)$. The remaining question is then to estimate $J(\mu_i)$. The episodic return $G(\tau_i)$ can be treated as a sample of $J(\mu_i)$. As a result, we can concentrate $J(\mu_{\mathcal{W}})$ near the weighted sum of returns with a direct application of Hoeffding's inequality (Serfling, 1974):

$$\mathbb{P}\left[\left|J(\mu_{\mathcal{W}}) - \sum_{i=0}^{N-1} w_i G(\tau_i)\right| > \epsilon\right] \leq 2 \exp\left(\frac{2\epsilon^2}{G_\top^2 \sum_{i=0}^{N-1} w_i^2}\right), \tag{7}$$

where $G_\top \doteq G_{\text{MAX}} - G_{\text{MIN}}$ is the return interval amplitude (see Hoeffding's inequality). For completeness, the soundness of the method is proved for any policy and MDP with discount factor (Sutton & Barto, 2018) less than 1 in Appendix A.1. Equation 7 tells us that we have a consistent estimator for $J(\mu_{\mathcal{W}})$ as long as too much mass has not been assigned to a small set of trajectories.

Since our goal is to obtain a behavior policy with a higher performance, we would like to give high weights $w_i$ to high performance $\mu_i$. However, it is worth noting that setting $w_i$ as a function of $G_i$ could induce a bias in the estimator of $J(\mu_W)$ due to the stochasticity in the trajectory generation, stemming from $\rho_0$, $P$, and/or $\mu_i$. In that case, Equation 7 concentration bound would not be valid anymore. To demonstrate and illustrate the bias, we provide a counterexample in Appendix A.3. The following section addresses this issue by making the strong assumption of determinism of the environment, and applying a trick to remove the stochasticity from the behavior policy $\mu_i$. Section 4.3 then relaxes the requirement for $\rho_0$ to be deterministic by using the return advantage instead of the absolute return.

## 4.2 RETURN-WEIGHTING

In this section, we are making the strong assumption that the MDP is deterministic (i.e., the transition dynamics $P$ and the initial state distribution $\rho_0$ is a Dirac delta distribution). This assumption allows us to obtain that $G(\tau_i) = J(\mu_i)$, where $\mu_i$ is the deterministic policy taking the actions in trajectory $\tau_i$[1]. Since the performance of the target policy is anchored on the performance of a behavior policy, we find a weighting distribution $\mathcal{W}$ to maximize $J(\mu_{\mathcal{W}})$:

$$\max_{\mathcal{W} \in \Delta_N} \sum_{i=0}^{N-1} w_i G(\tau_i), \tag{8}$$

where $w_i$ corresponds to the unnormalized weight assigned to each transition in episode $i$. However, the resulting solution is trivially assigning all the weights to the transitions in episode $\tau_i$ with maximum return. This trivial solution would indeed be optimal in the deterministic setting we consider but would fail otherwise. To prevent this from happening, we classically incorporate entropy regularization and turn Equation 8 into:

$$\max_{\mathcal{W} \in \Delta_N} \sum_{i=0}^{N-1} w_i G(\tau_i) - \alpha \sum_{i=0}^{N-1} w_i \log w_i, \tag{9}$$

---

[1]The formalization of $\mu_i$ is provided in Appendix A.1

where $\alpha \in \mathbb{R}^+$ is a temperature parameter that controls the strength of regularization. $\alpha$ interpolates the solution from a delta distribution ($\alpha \to 0$) to an uniform distribution ($\alpha \to \infty$). As the optimal solution to Equation 9 is a Boltzmann distribution of $G(\tau_i)$, the resulting weighting distribution $\mathcal{W}$ is:

$$w_i = \frac{\exp G(\tau_i)/\alpha}{\sum_{\tau_i \in \mathcal{D}} \exp G(\tau_i)/\alpha}. \tag{10}$$

The temperature parameter $\alpha$ (Equations 10 and 11) is a fixed hyperparameter. As the choice of $\alpha$ is dependent on the scale of episodic returns, which varies across environments, we normalize $G(\tau_i)$ using a max-min normalization: $G(\tau_i) \leftarrow \frac{G(\tau_i) - \min_j G(\tau_j)}{\max_j G(\tau_j) - \min_j G(\tau_j)}$.

## 4.3 ADVANTAGE-WEIGHTING

In this section, we allow the initial state distribution $\rho_0$ to be stochastic. The return-weighting strategy in Section 4.2 could be biased toward some trajectories starting from *lucky* initial states that yield higher returns than other initial states. Thus, we change the objective of Equation 9 to maximizing the weighted episodic advantage $\sum_{w_i \in \mathcal{W}} w_i A(\tau_i)$ with entropy regularization. $A(\tau_i)$ denotes the episodic advantage of $\tau_i$ and is defined as $A(\tau_i) = G(\tau_i) - \hat{V}^\mu(s_{i,0})$. $\hat{V}^\mu(s_{i,0})$ is the estimated expected return of following $\mu$ starting from $s_{i,0}$, using regression: $\hat{V}^\mu \leftarrow \arg\min_V \mathbb{E}\big[(G(\tau_i) - V(s_{i,0}))^2 \mid \text{Uni}(\mathcal{D})\big]$. Substituting $G(\tau_i)$ with $A(\tau_i)$ in Equation 9 and solving for $\mathcal{W}$, we obtain the following weighting distribution:

$$w_i = \frac{\exp A(\tau_i)/\alpha}{\sum_{\tau_i \in \mathcal{D}} \exp A(\tau_i)/\alpha}, \quad A(\tau_i) = G(\tau_i) - \hat{V}^\mu(s_{i,0}). \tag{11}$$

## 5 EXPERIMENTS

Our experiments answer the following primary questions: (i) Do our methods enable offline RL algorithms to achieve better performance in datasets with sparse high-return trajectories? (ii) Does our method benefit from high RPSV? (iii) Can our method also perform well in regular datasets? (iv) Is our method robust to stochasticity in an MDP?

### 5.1 SETUP

**Implementation.** We implement our weighted-sampling strategy and baselines in the following offline RL algorithms: implicit Q-learning (IQL) (Kostrikov et al., 2022), conservative Q-learning (CQL) (Kumar et al., 2020b), TD3+BC (Fujimoto & Gu, 2021), and behavior cloning (BC). IQL, CQL, and TD3+BC were chosen to cover various approaches of offline RL, including one-step, pessimistic, and conservative algorithms. Note that though BC is an imitation learning algorithm, we include it since BC clones the behavior policy, which is the object we directly alter, and BC is also a common baseline in offline RL research (Kumar et al., 2020b; Kostrikov et al., 2022).

**Baselines.** We compare our weighted sampling against uniform sampling (denoted as *Uniform*), percentage filtering (Chen et al., 2021) (denoted as *Top-x%*), and half-sampling (denoted as *Half*). Percentage filtering only uses episodes with top-$x\%$ returns for training. We consider percentage filtering as a baseline since it similarly increases the expected return of the behavior policy by discarding some data. In the following, we compare our method against *Top-*$10\%$ since $10\%$ is the best configuration found in the hyperparameter search (Appendix A.11). Half-sampling samples half of transitions from high-return and low-return trajectories, respectively. *Half* is a simple workaround to avoid over-sampling low-return data in datasets consisting of only sparse high-return trajectories. Note that *Half* requires the extra assumption of separating a dataset into high-return and low-return partitions, while our methods do not need this. Our return-weighted and advantage-weighted strategies are denoted as *RW* and *AW*, respectively, for which we use the same hyperparameter $\alpha$ in all the environments (see Appendix A.7).

**Datasets and environments.** We evaluate the performance of each algorithm+sampler variant (i.e., the combination of an offline RL algorithm and a sampling strategy) in MuJoCo locomotion

Figure 2: Our *RW* and *AW* sampling strategies achieve higher returns (y-axis) than all baselines (color) on average consistently for all algorithms CQL, IQL, BC, and TD3+BC, at all datasets with varying high-return data ratios $\sigma\%$ (x-axis). Remarkably, our performances in four algorithms exceed or match the average returns (dashed lines) of these algorithms trained with uniform sampling in full expert datasets. The substantial performance gain over *Uniform* at low ratios ($1\% \leq \sigma\% \leq 10\%$) shows the advantage of our methods in datasets with sparse high-return trajectories.

environments of D4RL benchmarks (Fu et al., 2020) and stochastic classic control benchmarks. Each environment is regarded as an MDP and can have multiple datasets in a benchmark suite. The dataset choices are described in the respective sections of the experimental results. We evaluate our method in stochastic classic control to investigate if stochastic dynamics break our weighted sampling strategies. The implementation of stochastic dynamics is presented in Appendix A.6.

**Evaluation metric.** An algorithm+sampler variant is trained for one million batches of updates in five random seeds for each dataset and environment. Its performance is measured by the average normalized episodic return of running the trained policy over 20 episodes in the environment. As suggested in Fu et al. (2020), we normalize the performance using $(X - X_{Random})/(X_{Expert} - X_{Random})$ where $X$, $X_{Random}$, and $X_{Expert}$ denote the performance of an algorithm-sampler variant, the random policy, and the expert one, respectively.

## 5.2 RESULTS IN MIXED DATASETS WITH SPARSE HIGH-RETURN TRAJECTORIES

To answer whether our weighted sampling methods improve the performance of uniform sampling in datasets with sparse high-return trajectories, we create mixed datasets with varying ratios of high-return data. We test each algorithm+sampler variant in four MuJoCo locomotion environments and eight mixed datasets, and one non-mixed dataset for each environment. The mixed datasets are created by mixing $\sigma\%$ of either an `expert` or `medium` datasets (high-return) with $(1 - \sigma\%)$ of a `random` dataset (low-return), for four ratios, $\sigma \in \{1, 5, 10, 50\}$. The `expert`, `medium`, and `random` datasets are generated by an expert policy, a policy with $1/3$ of the expert policy performance, and a random policy, respectively. We test all the variants in those 32 mixed datasets and `random` dataset.

Figure 2 shows the mean normalized performance (y-axis) of each algorithm+sampler (color) variant at varying $\sigma$ (x-axis). Each algorithm+sampler variant's performance is measured in the interquartile mean (IQM) (also known as 25%-trimmed mean) of average return (see Section 5.1) since IQM is less susceptible to the outlier performance as suggested in Agarwal et al. (2021). Appendix A.8 details the evaluation protocol.

It can be seen that in Figure 2 our *RW* and *AW* strategies significantly outperform the baselines *Uniform*, *Top-10%*, and *Half* for all algorithms at at all expert/medium data ratio $\sigma\%$. Remarkably, our methods even exceed or match the performance of each algorithm trained in full expert datasets with uniform sampling (dashed lines). This implies that our methods enable offline RL algorithms to achieve expert level of performance by 5% to 10% of medium or expert trajectories. *Uniform* fails to exploit to the fullest of the datasets when high-performing trajectories are sparse (i.e., low $\sigma$). *Top-10%* slightly improves the performance, yet fails to *Uniform* in low ratios ($\sigma\% = 1\%$), which implies the best filltering percentage might be dataset-dependent. *Half* consistently improves *Uniform* slightly at all ratios, yet the amounts of performance gain are far below ours. Overall, these results suggest that up-weighting high-return trajectories in a dataset with low ratios of high-return data benefits performance while naively filtering out low-return episodes, as *Top-10%* does, does not consistently improve performance. Moreover, *AW* and *RW* do not show visible differences, likely because the initial state distribution is narrow in MuJoCo locomotion environments. We also include the average returns in each environment and dataset in Appendix A.13. In addition to average return, we also evaluate our methods in the probability of improvements (Agarwal et al., 2021) over uniform sampling and show statistically significant improvements in Appendix A.10.

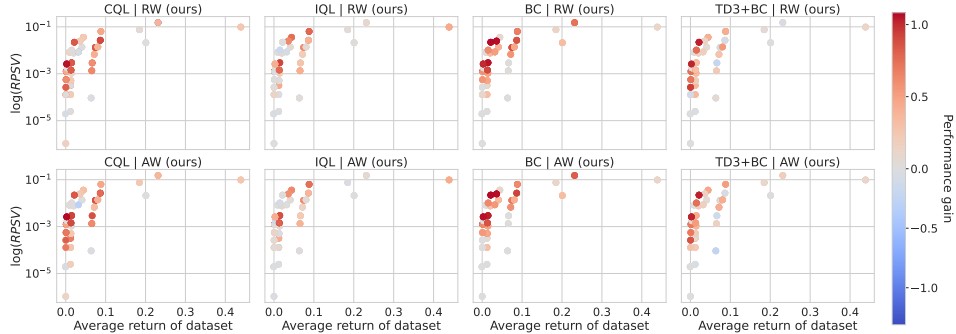

Figure 3: The left end (average return between 0 to 0.1) of each plot shows that for all offline RL algorithms (CQL, IQL, and TD3+BC) and BC, our AW and RW sampling methods' performance gain grows when RPSV increases. The color denotes the performance (average return) gain over the uniform sampling baseline in the mixed datasets and environments tested in Section 5.2; the x-axis and y-axis indicate the average return of a dataset and RPSV of the dataset, respectively.

## 5.3 ANALYSIS OF PERFORMANCE GAIN IN MIXED DATASETS

We hypothesize that our methods' performance gain over uniform sampling results from increased RPSV in the datasets. The design of a robust predictor for the performance gain of a sampling strategy is not trivial since offline RL's performance is influenced by several factors, including the environment and the offline RL algorithm that it is paired with. We focus on two statistical factors that are easy to estimate from the dataset: (i) the mean return of a dataset and (ii) RPSV. Although dependent on each other, these two factors have a good variability in our experiments since increasing the ratio of expert/medium data would increase not only RPSV but also the mean return of a dataset.

We show the relationship between the performance gain over uniform sampling (represented by the color of the dot in the plots below), datasets' mean return (x-axis), and RPSV (y-axis, in log scale) in Figure 3. Each dot denotes the average performance gain in a tuple of environment, dataset, and $\sigma$. It can be seen that at similar mean returns (x-axis), our methods' performance gain grows evidently (color gets closer to red) when RPSV increases (y-axis). This observation indicates that the performance gain with low $\sigma$ (expert/medium data ratio) in Figure 2 can be related to the performance gain at high RPSV since most datasets with low mean returns have high RPSV in our experiments. We also notice that a high dataset average return may temper our advantage. The reason is that offline RL with uniform sampling is already quite efficient in the settings where $\sigma$ is in a high range, such as 50%, and that the room for additional improvement over it is therefore limited.

## 5.4 RESULTS IN REGULAR DATASETS WITH MORE HIGH-RETURN TRAJECTORIES

Datasets in Section 5.2 are adversarially created to test the performance with extremely sparse high-return trajectories. However, we show in Figure 5 that such challenging return distributions are not common in regular datasets in D4RL benchmarks. As a result, regular datasets are easier than mixed datasets with sparse high-return trajectories for the uniform sampling baseline. To show that our method does not lose performance in regular datasets with more high-return trajectories, we also evaluate our method in 30 regular datasets from D4RL benchmark (Fu et al., 2020) using the same evaluation metric in Section 5.1, and present the results in Figure 4a. It can be seen that our methods both exhibit performance on par with the baselines in regular datasets, confirming that our method does not lose performance. Note that we do not compare with *Half* since regular datasets collected by multiple policies cannot be split into two buffers. Notably, we find that with our *RW* and *AW*, BC achieves competitive performance with other offline RL algorithms (i.e., CQL, IQL, and TD3+BC). The substantial improvement over uniform sampling in BC aligns with our analysis (Section 4.1) since the performance of BC solely depends on the performance of behavior policy and hence the average returns of sampled trajectories. Nonetheless, paired with *RW* and *AW*, offline RL algorithms (i.e., CQL, IQL, and TD3+BC) still outperform BC. This suggests that our weighted sampling strategies do not overshadow the advantage of offline RL over BC. The complete performance table can be found in Appendix A.13. We also evaluate our methods' probability of

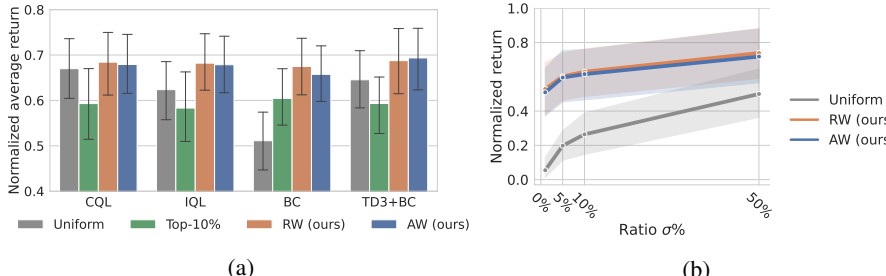

Figure 4: **(a)** Our method matches the *Uniform*'s return (y-axis). It indicates that our methods do not lose performance in datasets consisting of sufficient high-return trajectories. These datasets are regular datasets in D4RL without adversarially mixing low-return trajectories as we do in Section 5.2. **(b)** Performance in classic control tasks with stochastic dynamics. Our method outperforms the baselines, showing that stochasticity do not break our methods.

improvements (Agarwal et al., 2021) over uniform sampling, showing that our methods are no worse than baselines in Appendix A.8.1.

## 5.5 RESULTS IN STOCHASTIC MDPS

As our weighted-sampling strategy theoretically requires a deterministic MDP, we investigate if stochastic dynamics (*i.e.*, stochastic state transitions) break our method by evaluating it in stochastic control environments. The details of their implementation can be found in Appendix A.6. We use the evaluation metric described in Section 5.1 and present the results in Figure 4b. Both of our methods still outperform uniform sampling in stochastic dynamics, suggesting that stochasticity does not break them. Note that we only report the results with CQL since IQL and TD3+BC are not compatible with the discrete action space used in stochastic classic control.

## 6 RELATED WORKS

Our weighted sampling strategies and non-uniform experience replay in online RL aim to improve uniform sample selection. Prior works prioritize uncertain data (Schaul et al., 2015; Horgan et al., 2018; Lahire et al., 2021), attending on nearly on-policy samples (Sinha et al., 2022), or select samples by topological order Hong et al. (2022); Kumar et al. (2020a). However, these approaches do not take the performance of implicit behavioral policy induced by sampling into account and hence are unlikely to tackle the issue in mixed offline RL datasets.

Offline imitation learning (IL) (Kim et al., 2021; Ma et al., 2022; Xu et al., 2022) consider training an expert policy from a dataset consisting of a handful of expert data and plenty of random data. They train a model to discriminate if a transition is from an expert and learn a nearly expert policy from the discriminator's predictions. Conceptually, our methods and offline IL aim to capitalize advantageous data (i.e., sparse high-return/expert data) in a dataset despite different problem settings. Offline IL require that expert and random data are given in two separated buffer, but do not need reward labels. In contrast, we do not require separable datasets but require reward labels to find advantageous data.

## 7 DISCUSSION

**Importance of learning sparse high-return trajectories.** Though most regular datasets in mainstream offline RL benchmarks such as D4RL have more high-return trajectories than mixed datasets studied in Section 5.2, it should be noted that collecting these high-return data is tedious and could be expensive in realistic domains (e.g., health care). Thus, enabling offline RL to learn from datasets with a limited amount of high-return trajectories is crucial for deploying offline RL in more realistic tasks. The significance of our work is a simple technique to enable offline RL to learn from a handful of high-return trajectories.

**Limitation.** As our methods require trajectory returns to compute the sample weights, datasets cannot be partially fragmented trajectories, and each trajectory needs to start from states in the initial state distribution; otherwise, trajectory return cannot be estimated. One possible approach to lift this limitation is estimating the sample weight using a learned value function so that one can estimate the expected return of a state without complete trajectories.

## ACKNOWLEDGMENTS

We thank members of the Improbable AI Lab and Microsoft Research Montreal for helpful discussions and feedback. We are grateful to MIT Supercloud and the Lincoln Laboratory Supercomputing Center for providing HPC resources. This research was supported in part by the MIT-IBM Watson AI Lab, an AWS MLRA research grant, Google cloud credits provided as part of Google-MIT support, DARPA Machine Common Sense Program, ARO MURI under Grant Number W911NF-21-1-0328, ONR MURI under Grant Number N00014-22-1-2740, and by the United States Air Force Artificial Intelligence Accelerator under Cooperative Agreement Number FA8750-19-2-1000. The views and conclusions contained in this document are those of the authors and should not be interpreted as representing the official policies, either expressed or implied, of the Army Research Office or the United States Air Force or the U.S. Government. The U.S. Government is authorized to reproduce and distribute reprints for Government purposes notwithstanding any copyright notation herein.

## AUTHOR CONTRIBUTIONS

- **Zhang-Wei Hong** conceived the problem of mixed dataset in offline RL and the idea of trajectory reweighting, implemented the algorithms, ran the experiments, and wrote the paper.
- **Pulkit Agrawal** provided feedback on the idea and experiment designs.
- **Rémi Tachet des Combes** provided feedback on the idea and experiment designs, came up with the examples in Appendix A.3, and revised the paper.
- **Romain Laroche** formulated the analysis in Section 4, came up with RPSV metrics, and formulated the idea and revised the paper.

## REPRODUCIBILITY STATEMENT

We have included the implementation details in Appendix A.7 and the source code in the supplementary material.

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

# A  APPENDIX

## A.1  DETAILED ANALYSIS

We consider the definitions of a policy and a Markovian policy (Laroche et al., 2022):

**Definition 2** (Policy). *A policy $\pi$ represents any function mapping its trajectory history $h_t = \langle s_0, a_0, r_0 \ldots, s_{t-1}, a_{t-1}, r_{t-1}, s_t \rangle$ to a distribution over actions $\pi(\cdot|h_t) \in \Delta_{\mathcal{A}}$, where $\Delta_{\mathcal{A}}$ denotes the simplex over $\mathcal{A}$. Let $\Pi$ denote the space of policies, and $\Pi_{\mathrm{D}}$ the space of deterministic policies.*

**Definition 3** (Markovian policy). *Policy $\pi$ is said to be Markovian if its action probabilities only depend on the current state $s_t$: $\pi(\cdot|h_t) = \pi(\cdot|s_t) \in \Delta_{\mathcal{A}}$. Otherwise, policy $\pi$ is non-Markovian. We let $\Pi_{\mathrm{M}}$ denote the space of Markovian policies, and $\Pi_{\mathrm{DM}}$ the space of deterministic Markovian policies.*

We make no assumption on the behavior policy $\beta$, *i.e.* $\beta \in \Pi$. We notice that:

$$\mathcal{J}(\beta) = \mathbb{E}\left[R_\tau \mid \tau \sim p_0, \beta, p\right] = \mathbb{E}\left[R_\tau \mid \beta_\tau \sim \beta, \tau \sim p_0, \beta_\tau, p\right] = \mathbb{E}\left[\mathcal{J}(\beta_\tau) \mid \beta_\tau \sim \beta\right]. \quad (12)$$

Equation 12 is a trick that has already been used in Peng et al. (2019). We go a bit further by constraining $\beta_\tau$ to be a deterministic policy sampled at the start of the episode, which may be programmatically interpreted as sampling the random seed used for the full trajectory. With a trajectory-wise reweighting $\mathcal{W}$, we obtain:

$$\mathcal{J}(\beta_{\mathcal{W}}) = \sum_{i=0}^{N-1} w_i \mathcal{J}(\beta_{\tau_i}). \quad (13)$$

Furthermore, Altman (1999) tell us that there exists a Markovian policy $\beta_{\mathcal{W}}^{\mathrm{M}}$ with the same occupancy measure as $\beta_{\mathcal{W}}$ and the same performance when $\gamma < 1$ in MDPs with countable state space. Laroche et al. (2022) generalize this theorem to any MDP (including on uncountable state spaces) as long as $\gamma < 1$. Simão et al. (2020) prove in Theorem 3.2 Eq. (12) that, in finite MDPs, any Markovian behavior policy $\beta_{\mathcal{W}}^{\mathrm{M}}$ can be cloned with policy $\hat{\beta}_{\mathcal{W}}^{\mathrm{M}}$ from a dataset of $N$ trajectory up to accuracy of $\frac{2r_\top}{1-\gamma}\sqrt{\frac{3|\mathcal{S}||\mathcal{A}|+4\log\frac{1}{\delta}}{2N^2\sum_{i=0}^{N-1}w_i^2}}$[2] with high probability $1 - \delta$, where $2r_\top$ is the reward function amplitude.

## A.2  ADDITIONAL RELATED WORKS: IMBALANCE CLASSIFICATION/REGRESSION

Mixed datasets with high RPSV are closely related to imbalanced datasets in supervised learning. Supervised learning approaches either over-sample minority classes (Cui et al., 2019; Cao et al., 2019; Dong et al., 2018) or sample data inversely proportional to the target value densities (Yang et al., 2021; Steininger et al., 2021). Other works (Chawla et al., 2002; García & Herrera, 2009) synthesize samples by interpolating data points nearby the minority data. In RL, on the other hand, over-sampling minority data (trajectories) can be harmful if the trajectory is low-return and does not cover high-performing policies' trajectories; in other words, naive application of over-sampling techniques from supervised learning can hurt in an RL setting as they are agnostic to the notion of return.

## A.3  EXAMPLE OF BIAS WITH WEIGHTS THAT DEPEND ON THE REALIZATION OF THE TRAJECTORY

We will consider a minimal example consisting of a stateless MDP (multi-arm bandit) and 2 actions $\mathcal{A} = \{a_1, a_2\}$. Action $a_1$ yields a deterministic reward of 0.6. Action $a_2$ Bernouilli distribution reward with parameter $p = 0.5$. In other words, $a_2$ is a coin flip: with 50% chance, no reward is received and with 50% chance, a maximal reward of 1. We collect a dataset containing some number of samples for each action. Now consider weights $w_i$ such that:

$$w_i = \frac{\mathbb{1}[G(\tau_i) > 0.8]}{\sum_j \mathbb{1}[G(\tau_j) > 0.8]}. \quad (14)$$

---

[2]We implicitly replace in the denominator their unweigthed term $N$ with its weigthed version $N^2 \sum_{i=0}^{N-1} w_i^2$.

Then,

$$\mathcal{J}(\mu_{\mathcal{W}}) = \sum_{i=0}^{N-1} w_i \mathcal{J}(\mu_i) = \mathcal{J}(\mu(a_2) = 1) = 0.5 \tag{15}$$

$$\sum_{i=0}^{N-1} w_i G(\tau_i) = \sum_{i=0}^{N-1} \frac{\mathbb{1}[G(\tau_i) > 0.8]}{\sum_j \mathbb{1}[G(\tau_j) > 0.8]} = 1, \tag{16}$$

showing a counter-example for the concentration bound proposed in equation 7.

### A.4 DATASET AVERAGE RETURN

We plot the average return of mixed and regular datasets in Figure 5. It can be seen that mixed datasets used in Section 5.2 have lower average return than regular datasets on average. Also, we study the relationship between average return of dataset and offline RL performance in Figure 6, showing that increasing average return of dataset improves offline RL performance. Interestingly, there is a sweet spot where increasing datasets' average return starts hurting the performance. We hypothesize that it is due to insufficient state-aciton coverage of datasets. We also present the return distribution for each mixed dataset in Figure 7.

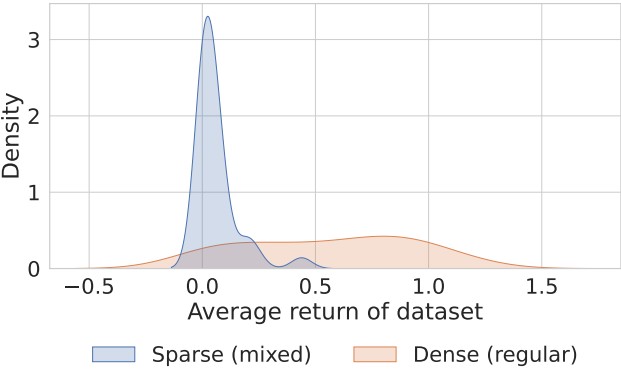

Figure 5: Average return of datasets. We see that mixed datasets used in Section 5.2 have lower average return than regular datasets on average.

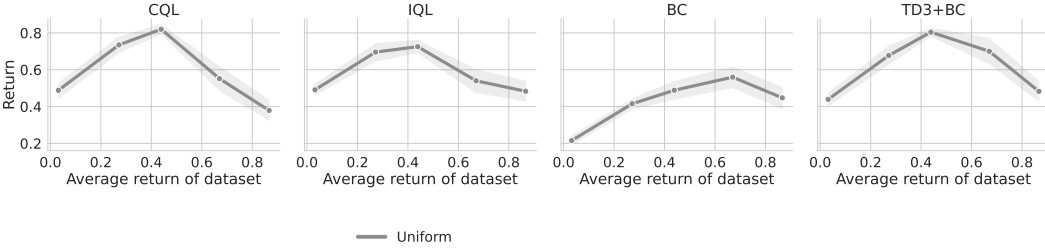

Figure 6: The relationship between average return of dataset (x-axis) and the performance (y-axis) of uniform sampling with an offline RL algorithm.

### A.5 DATASET RPSV

We list the PSV of each dataset in Table 1.

### A.6 STOCHASTIC CLASSIC CONTROL

We adapt `CartPole-v1`, `Acrobot-v1`, and `MountainCar-v0` in classic control environments in Open AI gym (Brockman et al., 2016). For each timestep, an agent's actions has $10\%$ chance to be replaced with noisy action $a \sim \mathcal{A}$. As such, the transitions dynamics turns to be stochastic.

|  | RPSV |
|---|---|
| ant-expert-v2 ($\sigma = 1$) | 0.002886 |
| ant-expert-v2 ($\sigma = 5$) | 0.013152 |
| ant-expert-v2 ($\sigma = 10$) | 0.026764 |
| ant-expert-v2 ($\sigma = 50$) | 0.151138 |
| ant-expert-v2 | 0.015052 |
| ant-full-replay-v2 | 0.096415 |
| ant-medium-expert-v2 | 0.042611 |
| ant-medium-replay-v2 | 0.038834 |
| ant-medium-v2 ($\sigma = 1$) | 0.001334 |
| ant-medium-v2 ($\sigma = 5$) | 0.006703 |
| ant-medium-v2 ($\sigma = 10$) | 0.013846 |
| ant-medium-v2 ($\sigma = 50$) | 0.075325 |
| ant-medium-v2 | 0.020586 |
| ant-random-v2 | 0.000092 |
| antmaze-large-diverse-v0 | 0.015378 |
| antmaze-large-play-v0 | 0.004538 |
| antmaze-medium-diverse-v0 | 0.072896 |
| antmaze-medium-play-v0 | 0.006583 |
| antmaze-umaze-diverse-v0 | 0.029304 |
| antmaze-umaze-v0 | 0.016956 |
| halfcheetah-expert-v2 ($\sigma = 1$) | 0.008236 |
| halfcheetah-expert-v2 ($\sigma = 5$) | 0.035784 |
| halfcheetah-expert-v2 ($\sigma = 10$) | 0.063523 |
| halfcheetah-expert-v2 ($\sigma = 50$) | 0.097583 |
| halfcheetah-expert-v2 | 0.000130 |
| halfcheetah-full-replay-v2 | 0.006521 |
| halfcheetah-medium-expert-v2 | 0.028588 |
| halfcheetah-medium-replay-v2 | 0.005747 |
| halfcheetah-medium-v2 ($\sigma = 1$) | 0.001771 |
| halfcheetah-medium-v2 ($\sigma = 5$) | 0.007604 |
| halfcheetah-medium-v2 ($\sigma = 10$) | 0.013685 |
| halfcheetah-medium-v2 ($\sigma = 50$) | 0.021007 |
| halfcheetah-medium-v2 | 0.000110 |
| halfcheetah-random-v2 | 0.000019 |
| hopper-expert-v2 ($\sigma = 1$) | 0.000318 |
| hopper-expert-v2 ($\sigma = 5$) | 0.001436 |
| hopper-expert-v2 ($\sigma = 10$) | 0.002948 |
| hopper-expert-v2 ($\sigma = 50$) | 0.024691 |
| hopper-expert-v2 | 0.000771 |
| hopper-full-replay-v2 | 0.041835 |
| hopper-medium-expert-v2 | 0.064473 |
| hopper-medium-replay-v2 | 0.018824 |
| hopper-medium-v2 ($\sigma = 1$) | 0.000129 |
| hopper-medium-v2 ($\sigma = 5$) | 0.000506 |
| hopper-medium-v2 ($\sigma = 10$) | 0.001055 |
| hopper-medium-v2 ($\sigma = 50$) | 0.008475 |
| hopper-medium-v2 | 0.006648 |
| hopper-random-v2 | 0.000025 |
| walker2d-expert-v2 ($\sigma = 1$) | 0.000264 |
| walker2d-expert-v2 ($\sigma = 5$) | 0.001261 |
| walker2d-expert-v2 ($\sigma = 10$) | 0.002601 |
| walker2d-expert-v2 ($\sigma = 50$) | 0.022037 |
| walker2d-expert-v2 | 0.000030 |
| walker2d-full-replay-v2 | 0.070379 |
| walker2d-medium-expert-v2 | 0.027597 |
| walker2d-medium-replay-v2 | 0.029698 |
| walker2d-medium-v2 ($\sigma = 1$) | 0.000128 |
| walker2d-medium-v2 ($\sigma = 5$) | 0.000559 |
| walker2d-medium-v2 ($\sigma = 10$) | 0.001233 |
| walker2d-medium-v2 ($\sigma = 50$) | 0.010165 |
| walker2d-medium-v2 | 0.018411 |
| walker2d-random-v2 | 0.000001 |

Table 1: RPSV calculated using normalized return.

## A.7    DETAILS OF IMPLEMENTATION

- **Temperature** $\alpha$**.** For RW and AW, we use $\alpha = 0.1$ for IQL and TD3+BC, and $\alpha = 0.2$ for CQL.
- **Trajectory advantage.** We use linear regression to approximate $V^\mu$. We make a training set $(s_{i,0}, G(\tau_i)) \ \forall \tau_i \in \mathcal{D}$, and train a regression model on the training set.
- **Implementation.** We use the public codebase, `d3rlpy` (Takuma Seno, 2021). For each algorithm, we use the hyperparamters as it is.

## A.8    DETAILS OF EVALUATION PROTOCOL

**Performance logging.**    Given an environment $E$ and a dataset $D_E$, for each trial (i.e., a random seed) we train each algorithm+sampler variant for one million batches using $D_E$ and rollout the learned policy in the environment $E$ for 20 episodes. The average return of the 20 episodes are booked as the performance of the trial.

**Performance metric.**    Given a list of empirical returns of eacy trial $[g_1, g_2, \cdots]$, interquantile mean (IQM) (Agarwal et al., 2021) discards the bottom $25\%$ and top $25\%$ samples and calculate the mean.

### A.8.1    PROBABILITY OF IMPROVEMENT

According to Agarwal et al. (2021), the probability of improvement in an environment $m$ is defined as:

$$P(X_m \geq Y_m) = \frac{1}{N^2} \sum_{i=1}^{N} \sum_{j=1}^{N} S(x_{m,i}, y_{m,j}), \ \ S(x_{m,i}, y_{m,j}) = \begin{cases} 1, x_{m,i} > y_{m,j} \\ \frac{1}{2}, x_{m,i} = y_{m,j} \\ 0, x_{m,i} < y_{m,j}, \end{cases}$$

where $m$ denote an environment index, $x_{m,i}$ and $y_{m,j}$ denote the samples of mean of return in trials of algorithms $X$ and $Y$, respectively. We report the average probability of improvements $\frac{1}{M} \sum_{m=0}^{M-1} P(X_m \geq Y_m)$ and its 95%-confidence interval using bootstrapping.

$PI$ cannot be directly translated into "number of winning" since $PI$ takes the stochasticity resulting from random seeds into account, measuring the probability of improvements in a trial with a randomly selected seed, dataset, and environment. For example, in Figure 9, we show that our methods attain 70% probability of improvements over uniform sampling, while this does not mean we beat uniform sampling in 70% of datasets and environments. From the complete score table in Appendix A.13, we see that our AW and RW strategies outperform uniform sampling in at least 80% of datasets.

We want to highlight that probability of improvement (PI) measures the robustness of a method, conveying different messages than the average performance shown in Figure 2 and Figure 4a. PI measures "how likely is a method to perform better than uniform sampling in a randomly selected environment, dataset, and random seed?" PI captures the uncertainty among random seeds while aggregated metrics like average performance does not. For example, suppose we have 5 trials with different random seeds on the same environment and dataset for two methods A and B. The fact that A has a higher average return than B, does not follow that A always performs better than B in all trials. It is possible that A is worse than B in some trials. Comparing only the average return, one would mis-conclude that A is certainly better than B. Instead, PI answers "how likely is A to be better than B?"

PI is important for algorithm selection since it measures the robustness of a method. One can have extremely a high performance gain in a few tasks and lose to baselines in the majority of tasks. If so, this new method would not always outperform baselines, which makes it not robust. A robust method should consistently improve baselines and not lose performance in most tasks.

Robustness of a method is important for a user to decide whether or not to prefer the new method over the existing method (i.e., baselines). As offline RL algorithms' performance interplay with several factors (e.g., dataset properties, environment dynamics, reward functions, etc), it is unlikely to accurately predict what conditions make the new method perform the best. Lacking of perfect knowledge of the best condition for the new method, it is unclear whether a user should deploy the

new method on a new task that does not have benchmarking results yet. As a result, robustness is crucial when selecting an algorithm for a new task. If the new method is shown to be robust and perform better than the baseline in most trials (i.e., high PI), it would be worth preferring the new method over the baseline. In contrast, it's not worth using the new method if the new method has PI below 50

### A.9 SENSITIVITY TO TEMPERATURE

The temperature $\alpha$ (Section 4) is an important hyperparameter of our method. We investigate how sensitive the choice of temperature algorithms is. Using the evaluation metric shown in Section 5.1, we compare the performance of *RW* and *AW* paired with IQL at varying temperature $\alpha$ in Figure 8, where $0.1$ is the temperature used in Sections 5.2, 5.4, and 5.5. Our methods outperform uniform sampling in a range of temperatures and hence are not overly sensitive to temperature. The full results in other offline algorithms are presented in Appendix A.12.

### A.10 PROBABILITY OF IMPROVEMENTS

In addition to average performance, the recent study by Agarwal et al. (2021) highlights the importance of measuring the robustness of an algorithm by its probability of improvement ($PI$) since outliers could dominate the average performance. An algorithm with a higher average performance does not necessarily perform better than baselines in the majority of environments. Therefore, we evaluate our method in both regular (Section 5.4) and mixed (Section 5.2) datasets in D4RL using probability of performing better than uniform sampling: $PI(X > Uniform)$, where $X \in \{Half, Top\text{-}10\%, RW, AW\}$. The bottom row of Figure 9 shows that our method achieves above 70% chance of outperforming uniform sampling in mixed datasets with sparse high-return trajectories. Moreover, the lower bounds of the confidence interval are clearly above 50%, which indicates that the improvements are significant according to Agarwal et al. (2021). On the other hand, in regular datasets with abundant high-return data, $PI(AW > Uniform)$ and $PI(RW > Uniform)$ are around 50%, suggesting that our methods match uniform sampling baseline. Note that $PI(RW > Uniform) = 70\%$ does not imply our method only beats uniform sampling in 70% of datasets and environments. The calculation of $PI$ is detailed in Appendix A.8.1.

### A.11 ADDITIONAL RESULTS OF PERCENTAGE FILTERING

Figure 10 presents the additional results at varying percentage for percentage filtering.

### A.12 ADDITIONAL RESULTS OF TEMPERATURE SENSITIVITY

Figure 11 presents the full results at varying temperatures.

### A.13 FULL RESULTS

We list the full benchmark results in Tables 2, 3, 4, and 5, where **bold** text denotes a score higher than *Uniform* and "*" sign indicates the maximum score in a dataset and environment (row).

### A.14 RESULTS IN OFFICIAL IQL CODEBASE

As the performance of our IQL implementation slightly mismatches the official implementation[3], we run the experiments in Sections 5.2 and 5.4 based on the official codebase and report the results in Figure 12. It can be seen that our methods still exhibit similar amounts of performance gain shown in Figure 2 and Figure 4a, indicating that the performance gain of our methods are independent of implementation.

---

[3] https://github.com/ikostrikov/implicit_q_learning

| Dataset | AW (ours) | RW (ours) | Top-10% | Uniform |
|---|---|---|---|---|
| ant-expert-v2 ($\sigma = 1$) | **0.31** | **0.31*** | 0.05 | 0.30 |
| ant-medium-v2 ($\sigma = 1$) | **0.34** | **0.35*** | 0.06 | 0.31 |
| halfcheetah-expert-v2 ($\sigma = 1$) | **0.02** | **0.03*** | 0.01 | 0.02 |
| halfcheetah-medium-v2 ($\sigma = 1$) | **0.02** | **0.03*** | 0.02 | 0.02 |
| hopper-expert-v2 ($\sigma = 1$) | **0.17*** | **0.16** | 0.03 | 0.04 |
| hopper-medium-v2 ($\sigma = 1$) | **0.22** | **0.34*** | 0.04 | 0.04 |
| walker2d-expert-v2 ($\sigma = 1$) | **0.02** | **0.02*** | **0.01** | 0.01 |
| walker2d-medium-v2 ($\sigma = 1$) | **0.03** | **0.05*** | **0.02** | 0.01 |
| ant-expert-v2 ($\sigma = 5$) | **1.01** | **1.08*** | 0.04 | 0.31 |
| ant-medium-v2 ($\sigma = 5$) | **0.86*** | **0.86** | 0.10 | 0.31 |
| halfcheetah-expert-v2 ($\sigma = 5$) | **0.14*** | **0.11** | 0.02 | 0.02 |
| halfcheetah-medium-v2 ($\sigma = 5$) | **0.36*** | **0.36** | **0.14** | 0.02 |
| hopper-expert-v2 ($\sigma = 5$) | **0.97*** | **0.87** | **0.05** | 0.04 |
| hopper-medium-v2 ($\sigma = 5$) | **0.54*** | **0.49** | **0.15** | 0.05 |
| walker2d-expert-v2 ($\sigma = 5$) | **1.03*** | **1.01** | **0.01** | 0.01 |
| walker2d-medium-v2 ($\sigma = 5$) | **0.60*** | **0.56** | **0.11** | 0.02 |
| ant-expert-v2 ($\sigma = 10$) | **1.19** | **1.19*** | 0.15 | 0.34 |
| ant-medium-v2 ($\sigma = 10$) | **0.85** | **0.91*** | 0.27 | 0.45 |
| halfcheetah-expert-v2 ($\sigma = 10$) | **0.68** | **0.76** | **0.77*** | 0.02 |
| halfcheetah-medium-v2 ($\sigma = 10$) | **0.40** | **0.42** | **0.42*** | 0.02 |
| hopper-expert-v2 ($\sigma = 10$) | **1.03** | **1.05*** | **0.06** | 0.05 |
| hopper-medium-v2 ($\sigma = 10$) | **0.57*** | **0.57** | **0.23** | 0.05 |
| walker2d-expert-v2 ($\sigma = 10$) | **1.08*** | **1.06** | 0.01 | 0.03 |
| walker2d-medium-v2 ($\sigma = 10$) | **0.68*** | **0.62** | **0.30** | 0.06 |
| ant-expert-v2 ($\sigma = 50$) | **1.24** | **1.21** | **1.26*** | 0.44 |
| ant-medium-v2 ($\sigma = 50$) | **0.88** | **0.89*** | **0.89** | 0.73 |
| halfcheetah-expert-v2 ($\sigma = 50$) | **0.92** | **0.92*** | 0.40 | 0.80 |
| halfcheetah-medium-v2 ($\sigma = 50$) | **0.42*** | **0.42** | **0.41** | 0.10 |
| hopper-expert-v2 ($\sigma = 50$) | **1.10*** | **1.10** | **0.86** | 0.04 |
| hopper-medium-v2 ($\sigma = 50$) | **0.56** | **0.57*** | **0.51** | 0.03 |
| walker2d-expert-v2 ($\sigma = 50$) | **1.08*** | **1.08** | **0.25** | 0.01 |
| walker2d-medium-v2 ($\sigma = 50$) | **0.71** | **0.72*** | **0.59** | 0.10 |
| ant-expert-v2 | **1.25*** | 1.24 | 1.23 | 1.25 |
| ant-full-replay-v2 | **1.26** | **1.28*** | **1.25** | 1.18 |
| ant-medium-expert-v2 | **1.25** | **1.27*** | **1.23** | 1.17 |
| ant-medium-replay-v2 | **0.80*** | **0.79** | 0.66 | 0.67 |
| ant-medium-v2 | **0.86** | **0.88** | **0.93*** | 0.86 |
| ant-random-v2 | 0.30 | 0.29 | 0.06 | 0.32* |
| antmaze-large-diverse-v0 | **0.12** | **0.16*** | 0.00 | 0.00 |
| antmaze-large-play-v0 | **0.09** | **0.13*** | 0.00 | 0.00 |
| antmaze-medium-diverse-v0 | **0.10** | **0.29*** | **0.01** | 0.00 |
| antmaze-medium-play-v0 | **0.13** | **0.21*** | **0.01** | 0.00 |
| antmaze-umaze-diverse-v0 | **0.58** | **0.65*** | 0.51 | 0.54 |
| antmaze-umaze-v0 | **0.59** | **0.60** | **0.65*** | 0.49 |
| halfcheetah-expert-v2 | **0.92*** | **0.92** | 0.70 | 0.92 |
| halfcheetah-full-replay-v2 | **0.67** | **0.67*** | **0.64** | 0.62 |
| halfcheetah-medium-expert-v2 | **0.92** | **0.92*** | **0.88** | 0.58 |
| halfcheetah-medium-replay-v2 | **0.38** | **0.40*** | 0.31 | 0.34 |
| halfcheetah-medium-v2 | 0.42 | 0.42 | 0.42 | 0.43* |
| halfcheetah-random-v2 | 0.02 | 0.02 | 0.02 | 0.02* |
| hopper-expert-v2 | 1.08 | **1.11*** | 1.06 | 1.09 |
| hopper-full-replay-v2 | **0.98** | **1.01*** | **0.93** | 0.31 |
| hopper-medium-expert-v2 | **1.10** | **1.10*** | **1.09** | 0.53 |
| hopper-medium-replay-v2 | **0.72*** | **0.67** | **0.63** | 0.27 |
| hopper-medium-v2 | **0.56** | **0.54** | **0.58*** | 0.52 |
| hopper-random-v2 | **0.05*** | **0.05** | **0.05** | 0.04 |
| walker2d-expert-v2 | **1.09*** | **1.08** | 1.03 | 1.08 |
| walker2d-full-replay-v2 | **0.84** | **0.84** | **0.88*** | 0.27 |
| walker2d-medium-expert-v2 | **1.08** | **1.08*** | **1.08** | 0.97 |
| walker2d-medium-replay-v2 | **0.56** | **0.56*** | **0.51** | 0.19 |
| walker2d-medium-v2 | **0.70** | **0.71*** | **0.70** | 0.64 |
| walker2d-random-v2 | **0.01** | **0.01*** | **0.01** | 0.01 |
| Num. win Uniform | 58 | 58 | 39 | - |

Table 2: BC results

| Dataset | AW (ours) | RW (ours) | Top-10% | Uniform |
|---|---|---|---|---|
| ant-expert-v2 ($\sigma = 1$) | **0.54*** | **0.54** | 0.06 | 0.25 |
| ant-medium-v2 ($\sigma = 1$) | **0.82** | **0.82*** | 0.07 | 0.39 |
| halfcheetah-expert-v2 ($\sigma = 1$) | 0.03 | **0.04*** | 0.03 | 0.03 |
| halfcheetah-medium-v2 ($\sigma = 1$) | 0.16 | 0.20 | 0.07 | 0.21* |
| hopper-expert-v2 ($\sigma = 1$) | **0.55*** | **0.50** | 0.11 | 0.12 |
| hopper-medium-v2 ($\sigma = 1$) | 0.39 | 0.50 | 0.30 | 0.53* |
| walker2d-expert-v2 ($\sigma = 1$) | **0.29*** | **0.28** | 0.10 | 0.12 |
| walker2d-medium-v2 ($\sigma = 1$) | **0.46** | **0.48*** | 0.12 | 0.36 |
| ant-expert-v2 ($\sigma = 5$) | **1.16** | **1.19*** | 0.12 | 0.60 |
| ant-medium-v2 ($\sigma = 5$) | **0.88** | **0.91*** | 0.28 | 0.73 |
| halfcheetah-expert-v2 ($\sigma = 5$) | **0.66** | **0.71*** | 0.04 | 0.04 |
| halfcheetah-medium-v2 ($\sigma = 5$) | **0.43*** | **0.41** | 0.30 | 0.33 |
| hopper-expert-v2 ($\sigma = 5$) | **1.02*** | **0.87** | **0.18** | 0.12 |
| hopper-medium-v2 ($\sigma = 5$) | **0.53** | **0.56*** | 0.48 | 0.49 |
| walker2d-expert-v2 ($\sigma = 5$) | **1.08** | **1.08*** | **0.53** | 0.25 |
| walker2d-medium-v2 ($\sigma = 5$) | 0.58 | **0.61*** | 0.49 | 0.58 |
| ant-expert-v2 ($\sigma = 10$) | **1.18** | **1.23*** | 0.40 | 0.74 |
| ant-medium-v2 ($\sigma = 10$) | **0.87** | **0.90*** | 0.62 | 0.79 |
| halfcheetah-expert-v2 ($\sigma = 10$) | **0.85** | **0.89*** | **0.85** | 0.08 |
| halfcheetah-medium-v2 ($\sigma = 10$) | **0.45** | **0.45** | **0.45*** | 0.40 |
| hopper-expert-v2 ($\sigma = 10$) | **1.00** | **1.01*** | 0.36 | 0.18 |
| hopper-medium-v2 ($\sigma = 10$) | **0.55** | **0.60*** | 0.54 | 0.46 |
| walker2d-expert-v2 ($\sigma = 10$) | **1.09*** | **1.08** | **0.92** | 0.66 |
| walker2d-medium-v2 ($\sigma = 10$) | **0.67*** | 0.64 | **0.66** | 0.65 |
| ant-expert-v2 ($\sigma = 50$) | **1.21** | **1.21** | **1.24*** | 1.07 |
| ant-medium-v2 ($\sigma = 50$) | **0.92** | **0.94** | **0.96*** | 0.91 |
| halfcheetah-expert-v2 ($\sigma = 50$) | **0.91** | **0.94*** | **0.70** | 0.50 |
| halfcheetah-medium-v2 ($\sigma = 50$) | **0.47** | **0.47*** | 0.44 | 0.45 |
| hopper-expert-v2 ($\sigma = 50$) | **1.06** | **1.07*** | 0.43 | 0.50 |
| hopper-medium-v2 ($\sigma = 50$) | 0.54 | 0.52 | **0.62*** | 0.56 |
| walker2d-expert-v2 ($\sigma = 50$) | **1.09*** | **1.09** | 1.08 | 1.08 |
| walker2d-medium-v2 ($\sigma = 50$) | 0.63 | 0.63 | **0.71*** | 0.67 |
| ant-expert-v2 | **1.25** | **1.26** | **1.27*** | 1.13 |
| ant-full-replay-v2 | **1.29*** | **1.27** | **1.25** | 1.24 |
| ant-medium-expert-v2 | **1.23** | **1.32*** | **1.28** | 1.14 |
| ant-medium-replay-v2 | 0.83 | **0.86*** | 0.72 | 0.83 |
| ant-medium-v2 | 0.97 | 0.98 | 0.88 | 0.99* |
| ant-random-v2 | **0.14*** | **0.14** | 0.06 | 0.12 |
| antmaze-large-diverse-v0 | **0.40*** | 0.24 | **0.04** | 0.00 |
| antmaze-large-play-v0 | 0.18 | **0.33*** | 0.00 | 0.00 |
| antmaze-medium-diverse-v0 | 0.24 | **0.31*** | 0.02 | 0.05 |
| antmaze-medium-play-v0 | 0.41 | **0.43*** | **0.03** | 0.03 |
| antmaze-umaze-diverse-v0 | 0.54 | 0.40 | **0.61*** | 0.46 |
| antmaze-umaze-v0 | **0.87*** | 0.86 | 0.72 | 0.85 |
| halfcheetah-expert-v2 | **0.94*** | 0.93 | 0.74 | 0.93 |
| halfcheetah-full-replay-v2 | **0.75*** | **0.75** | **0.71** | 0.70 |
| halfcheetah-medium-expert-v2 | **0.94*** | 0.93 | 0.89 | 0.71 |
| halfcheetah-medium-replay-v2 | **0.44*** | 0.44 | 0.35 | 0.44 |
| halfcheetah-medium-v2 | **0.47** | **0.47*** | 0.45 | 0.47 |
| halfcheetah-random-v2 | 0.07 | 0.07 | 0.04 | 0.12* |
| hopper-expert-v2 | **1.00*** | **0.96** | 0.92 | 0.95 |
| hopper-full-replay-v2 | **0.97*** | 0.79 | **0.97** | 0.89 |
| hopper-medium-expert-v2 | **0.99** | **1.01** | **1.05*** | 0.59 |
| hopper-medium-replay-v2 | **0.84** | **0.86*** | **0.85** | 0.83 |
| hopper-medium-v2 | **0.57** | **0.57** | **0.65*** | 0.55 |
| hopper-random-v2 | 0.06 | 0.07 | 0.08 | 0.09* |
| walker2d-expert-v2 | **1.09*** | **1.09** | **1.09** | 1.09 |
| walker2d-full-replay-v2 | 0.75 | 0.83 | 0.76 | 0.92* |
| walker2d-medium-expert-v2 | **1.10*** | **1.09** | **1.09** | 1.06 |
| walker2d-medium-replay-v2 | 0.47 | 0.37 | 0.51 | 0.69* |
| walker2d-medium-v2 | 0.69 | 0.66 | 0.65 | 0.75* |
| walker2d-random-v2 | 0.03 | 0.04 | **0.11*** | 0.06 |
| Num. win Uniform | 48 | 47 | 29 | - |

Table 3: IQL results

| Dataset | AW (ours) | RW (ours) | Top-10% | Uniform |
|---|---|---|---|---|
| ant-expert-v2 ($\sigma = 1$) | **0.80*** | **0.74** | 0.04 | 0.08 |
| ant-medium-v2 ($\sigma = 1$) | **0.79** | **0.80*** | 0.06 | 0.14 |
| halfcheetah-expert-v2 ($\sigma = 1$) | **0.39*** | **0.38** | 0.09 | 0.25 |
| halfcheetah-medium-v2 ($\sigma = 1$) | **0.41** | **0.42*** | 0.26 | 0.40 |
| hopper-expert-v2 ($\sigma = 1$) | **0.84*** | **0.52** | **0.15** | 0.09 |
| hopper-medium-v2 ($\sigma = 1$) | **0.61*** | **0.58** | **0.43** | 0.30 |
| walker2d-expert-v2 ($\sigma = 1$) | **0.57*** | **0.54** | 0.01 | 0.03 |
| walker2d-medium-v2 ($\sigma = 1$) | **0.46** | **0.48*** | **0.12** | 0.03 |
| ant-expert-v2 ($\sigma = 5$) | **1.06** | **1.06*** | **0.35** | 0.15 |
| ant-medium-v2 ($\sigma = 5$) | **0.88** | **0.93*** | **0.57** | 0.50 |
| halfcheetah-expert-v2 ($\sigma = 5$) | **0.67*** | **0.66** | 0.10 | 0.30 |
| halfcheetah-medium-v2 ($\sigma = 5$) | **0.47*** | 0.47 | 0.44 | 0.47 |
| hopper-expert-v2 ($\sigma = 5$) | **0.99*** | **0.99** | **0.31** | 0.10 |
| hopper-medium-v2 ($\sigma = 5$) | **0.65** | **0.68*** | **0.63** | 0.43 |
| walker2d-expert-v2 ($\sigma = 5$) | **1.06*** | **1.02** | 0.00 | 0.03 |
| walker2d-medium-v2 ($\sigma = 5$) | **0.76*** | **0.75** | **0.41** | 0.04 |
| ant-expert-v2 ($\sigma = 10$) | **1.11** | **1.16*** | **0.93** | 0.21 |
| ant-medium-v2 ($\sigma = 10$) | **0.91** | **0.96*** | **0.91** | 0.57 |
| halfcheetah-expert-v2 ($\sigma = 10$) | **0.76** | **0.76*** | **0.74** | 0.29 |
| halfcheetah-medium-v2 ($\sigma = 10$) | **0.48** | **0.48*** | 0.47 | 0.48 |
| hopper-expert-v2 ($\sigma = 10$) | **1.08*** | **1.03** | **0.37** | 0.14 |
| hopper-medium-v2 ($\sigma = 10$) | **0.68** | **0.71** | **0.72*** | 0.62 |
| walker2d-expert-v2 ($\sigma = 10$) | **1.02** | **1.09*** | **0.12** | 0.01 |
| walker2d-medium-v2 ($\sigma = 10$) | **0.79*** | **0.79** | **0.68** | 0.24 |
| ant-expert-v2 ($\sigma = 50$) | **1.26** | 0.53 | **1.27*** | 0.83 |
| ant-medium-v2 ($\sigma = 50$) | **0.97** | **0.94** | **0.98*** | 0.82 |
| halfcheetah-expert-v2 ($\sigma = 50$) | **0.80** | **0.86*** | 0.56 | 0.57 |
| halfcheetah-medium-v2 ($\sigma = 50$) | 0.49 | 0.49 | 0.47 | **0.49*** |
| hopper-expert-v2 ($\sigma = 50$) | **1.02** | **1.04** | **1.04*** | 0.82 |
| hopper-medium-v2 ($\sigma = 50$) | 0.63 | 0.38 | 0.72 | **0.73*** |
| walker2d-expert-v2 ($\sigma = 50$) | **1.09*** | **1.08** | **0.96** | 0.24 |
| walker2d-medium-v2 ($\sigma = 50$) | **0.82** | **0.83*** | **0.81** | 0.81 |
| ant-expert-v2 | **1.27*** | **1.27** | 1.16 | 1.21 |
| ant-full-replay-v2 | **1.27** | **1.24** | **1.31*** | 1.22 |
| ant-medium-expert-v2 | **1.28*** | **1.22** | **1.23** | 1.17 |
| ant-medium-replay-v2 | **0.91** | **0.97*** | 0.83 | 0.90 |
| ant-medium-v2 | 0.95 | 0.97 | 0.96 | **0.98*** |
| ant-random-v2 | 0.08 | 0.08 | 0.06 | **0.11*** |
| antmaze-large-diverse-v0 | 0.00 | 0.00 | **0.08*** | 0.00 |
| antmaze-large-play-v0 | 0.00 | **0.01*** | 0.00 | 0.00 |
| antmaze-medium-diverse-v0 | 0.00 | **0.08*** | 0.00 | 0.00 |
| antmaze-medium-play-v0 | 0.00 | **0.03*** | **0.01** | 0.00 |
| antmaze-umaze-diverse-v0 | **0.22** | **0.10** | **0.23*** | 0.05 |
| antmaze-umaze-v0 | 0.65 | **0.71*** | 0.25 | 0.65 |
| halfcheetah-expert-v2 | **0.89*** | **0.88** | 0.77 | 0.79 |
| halfcheetah-full-replay-v2 | **0.79** | **0.79*** | 0.77 | 0.78 |
| halfcheetah-medium-expert-v2 | **0.84** | **0.85*** | **0.70** | 0.63 |
| halfcheetah-medium-replay-v2 | 0.47 | 0.47 | 0.42 | **0.47*** |
| halfcheetah-medium-v2 | 0.49 | **0.49*** | 0.47 | 0.49 |
| halfcheetah-random-v2 | 0.17 | 0.17 | 0.03 | **0.23*** |
| hopper-expert-v2 | 1.05 | 1.03 | 1.01 | **1.05*** |
| hopper-full-replay-v2 | **1.03** | **1.07*** | **1.01** | 1.00 |
| hopper-medium-expert-v2 | 0.91 | **1.00** | **1.04*** | 0.99 |
| hopper-medium-replay-v2 | **0.99*** | **0.99** | **0.96** | 0.95 |
| hopper-medium-v2 | 0.71 | 0.73 | 0.71 | **0.74*** |
| hopper-random-v2 | **0.27*** | **0.24** | 0.08 | 0.12 |
| walker2d-expert-v2 | **1.09** | **1.09*** | 1.09 | 1.09 |
| walker2d-full-replay-v2 | 0.86 | 0.85 | 0.91 | **0.94*** |
| walker2d-medium-expert-v2 | **1.09** | **1.08** | **1.09*** | 1.08 |
| walker2d-medium-replay-v2 | **0.87*** | **0.86** | 0.71 | 0.84 |
| walker2d-medium-v2 | **0.83*** | 0.82 | 0.74 | 0.83 |
| walker2d-random-v2 | **0.14** | **0.14** | **0.17*** | 0.03 |
| Num. win Uniform | 50 | 50 | 33 | - |

Table 4: CQL results

| Dataset | AW (ours) | RW (ours) | Top-10% | Uniform |
|---|---|---|---|---|
| ant-expert-v2 ($\sigma = 1$) | 0.18 | 0.17 | 0.03 | 0.25* |
| ant-medium-v2 ($\sigma = 1$) | **0.48*** | **0.46** | 0.01 | 0.39 |
| halfcheetah-expert-v2 ($\sigma = 1$) | 0.04 | 0.04 | **0.08*** | 0.04 |
| halfcheetah-medium-v2 ($\sigma = 1$) | **0.21** | **0.18** | **0.31*** | 0.16 |
| hopper-expert-v2 ($\sigma = 1$) | **0.57*** | **0.52** | 0.12 | 0.21 |
| hopper-medium-v2 ($\sigma = 1$) | **0.53** | **0.54*** | 0.21 | 0.30 |
| walker2d-expert-v2 ($\sigma = 1$) | **0.69** | **0.90*** | 0.03 | 0.05 |
| walker2d-medium-v2 ($\sigma = 1$) | **0.18*** | **0.17** | 0.06 | 0.13 |
| ant-expert-v2 ($\sigma = 5$) | **0.44*** | **0.40** | 0.08 | 0.32 |
| ant-medium-v2 ($\sigma = 5$) | **0.99*** | **0.81** | 0.25 | 0.58 |
| halfcheetah-expert-v2 ($\sigma = 5$) | **0.59** | **0.62*** | 0.08 | 0.14 |
| halfcheetah-medium-v2 ($\sigma = 5$) | **0.47*** | **0.47** | **0.45** | 0.30 |
| hopper-expert-v2 ($\sigma = 5$) | **0.97*** | **0.88** | **0.44** | 0.42 |
| hopper-medium-v2 ($\sigma = 5$) | **0.55** | **0.55*** | **0.42** | 0.38 |
| walker2d-expert-v2 ($\sigma = 5$) | **0.66** | **0.70*** | 0.02 | 0.13 |
| walker2d-medium-v2 ($\sigma = 5$) | **0.64** | **0.78*** | 0.04 | 0.05 |
| ant-expert-v2 ($\sigma = 10$) | **0.53*** | **0.46** | 0.16 | 0.46 |
| ant-medium-v2 ($\sigma = 10$) | **1.09*** | **1.05** | 0.39 | 0.60 |
| halfcheetah-expert-v2 ($\sigma = 10$) | **0.79** | **0.82*** | **0.81** | 0.31 |
| halfcheetah-medium-v2 ($\sigma = 10$) | **0.48** | **0.47** | **0.48*** | 0.43 |
| hopper-expert-v2 ($\sigma = 10$) | **1.02** | **1.05*** | 0.64 | 0.65 |
| hopper-medium-v2 ($\sigma = 10$) | **0.57*** | **0.57** | **0.53** | 0.24 |
| walker2d-expert-v2 ($\sigma = 10$) | **1.10*** | **0.81** | 0.03 | 0.08 |
| walker2d-medium-v2 ($\sigma = 10$) | **0.76*** | **0.62** | 0.03 | 0.06 |
| ant-expert-v2 ($\sigma = 50$) | 0.48 | **0.57*** | 0.47 | 0.51 |
| ant-medium-v2 ($\sigma = 50$) | **1.11*** | **1.06** | **1.00** | 0.90 |
| halfcheetah-expert-v2 ($\sigma = 50$) | **0.94** | **0.95*** | 0.69 | 0.80 |
| halfcheetah-medium-v2 ($\sigma = 50$) | **0.48*** | **0.48** | 0.47 | 0.48 |
| hopper-expert-v2 ($\sigma = 50$) | **1.11*** | **1.10** | **1.01** | 0.97 |
| hopper-medium-v2 ($\sigma = 50$) | **0.62*** | **0.60** | **0.52** | 0.44 |
| walker2d-expert-v2 ($\sigma = 50$) | **1.10*** | **1.10** | 0.07 | 0.08 |
| walker2d-medium-v2 ($\sigma = 50$) | **0.80** | **0.81*** | **0.34** | 0.07 |
| ant-expert-v2 | **0.96** | **0.98*** | **0.83** | 0.55 |
| ant-full-replay-v2 | **1.34** | **1.38*** | 1.28 | 1.34 |
| ant-medium-expert-v2 | **1.13*** | **1.12** | **0.95** | 0.70 |
| ant-medium-replay-v2 | 0.96 | 1.00 | 0.64 | 1.06* |
| ant-medium-v2 | 1.17 | 1.16 | 0.86 | 1.20* |
| ant-random-v2 | 0.22 | 0.36 | -0.02 | 0.37* |
| antmaze-large-diverse-v0 | 0.00 | **0.01** | **0.03*** | 0.00 |
| antmaze-large-play-v0 | **0.00*** | **0.00*** | **0.00*** | **0.00*** |
| antmaze-medium-diverse-v0 | **0.10*** | **0.05** | **0.07** | 0.01 |
| antmaze-medium-play-v0 | **0.07*** | **0.01** | **0.04** | 0.00 |
| antmaze-umaze-diverse-v0 | **0.52*** | 0.26 | 0.17 | 0.47 |
| antmaze-umaze-v0 | 0.50 | 0.72 | 0.51 | 0.79* |
| halfcheetah-expert-v2 | **0.97*** | **0.97** | 0.84 | 0.95 |
| halfcheetah-full-replay-v2 | **0.78*** | **0.77** | **0.76** | 0.73 |
| halfcheetah-medium-expert-v2 | **0.97*** | **0.96** | 0.86 | 0.87 |
| halfcheetah-medium-replay-v2 | **0.44** | **0.45*** | 0.34 | 0.44 |
| halfcheetah-medium-v2 | **0.48*** | **0.48** | 0.48 | 0.48 |
| halfcheetah-random-v2 | 0.11 | **0.11** | **0.12*** | 0.11 |
| hopper-expert-v2 | 1.05 | 1.10 | 1.06 | 1.11* |
| hopper-full-replay-v2 | **1.04*** | **1.03** | **1.02** | 0.73 |
| hopper-medium-expert-v2 | **1.03** | **1.10*** | **1.10** | 0.96 |
| hopper-medium-replay-v2 | **0.93** | **0.95*** | **0.85** | 0.59 |
| hopper-medium-v2 | **0.62** | **0.62** | **0.64*** | 0.56 |
| hopper-random-v2 | 0.07 | 0.07 | 0.08 | 0.10* |
| walker2d-expert-v2 | **1.10*** | **1.10** | 1.10 | 1.10 |
| walker2d-full-replay-v2 | **0.96*** | **0.96** | 0.92 | 0.94 |
| walker2d-medium-expert-v2 | **1.10** | **1.10** | **1.10*** | 1.10 |
| walker2d-medium-replay-v2 | **0.80*** | 0.75 | 0.78 | 0.80 |
| walker2d-medium-v2 | 0.80 | 0.81 | 0.67 | 0.82* |
| walker2d-random-v2 | 0.03 | 0.02 | 0.02 | 0.04* |
| Num. win Uniform | 49 | 48 | 25 | - |

Table 5: TD3+BC results

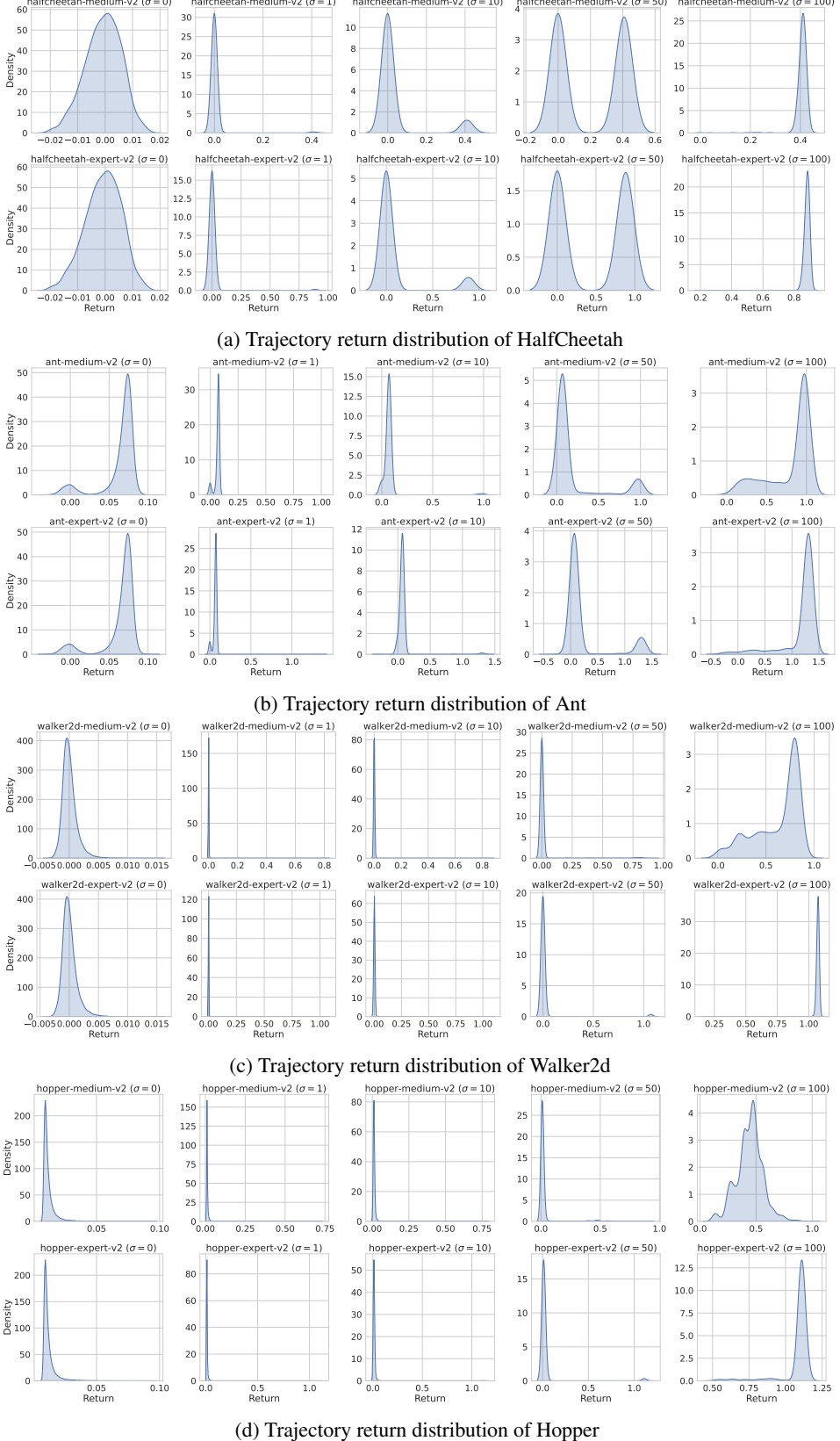

(a) Trajectory return distribution of HalfCheetah

(b) Trajectory return distribution of Ant

(c) Trajectory return distribution of Walker2d

(d) Trajectory return distribution of Hopper

Figure 7: Trajectory return distribution of each mixed dataset.

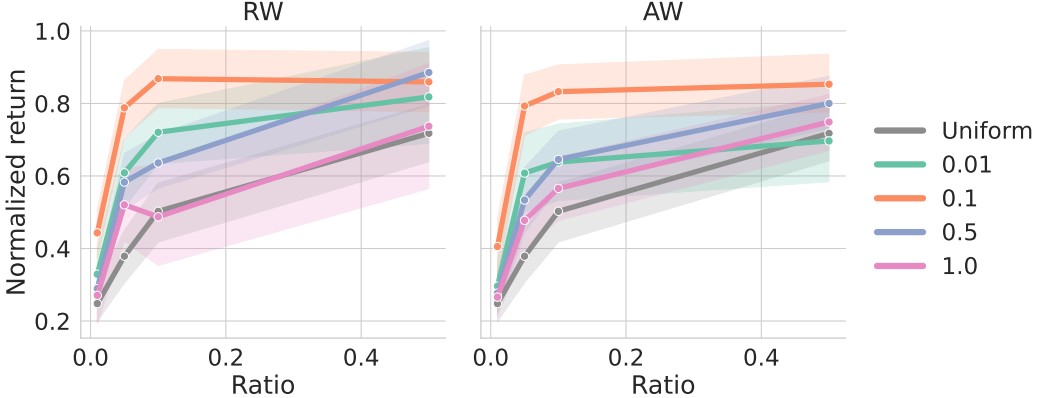

Figure 8: Performance of our method with varying temperature $\alpha$ (Sections 4.2 and 4.3), where color denotes $\alpha$. Both *AW* and *RW* achieve higher returns than the baselines in a wide range of temperatures $[\![0.01, 1.0]\!]$, our methods are not overly sensitive to the choice of temperature.

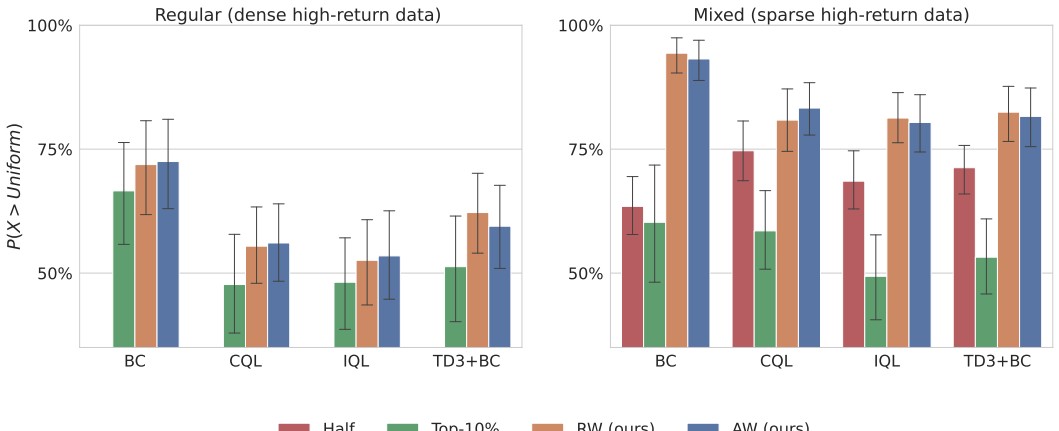

Figure 9: Probability of improvement over uniform sampling (Agarwal et al., 2021) in 32 mixed (lower row) and 30 regular datasets (upper row). In mixed datasets with sparse high-return data, our methods attains above 75% of the probability of improvements with a lower bound of confidence interval clearly above 50%, suggesting statistically significant improvements over uniform sampling. On the other hand, in regular datasets with abundant high-return data, $PI(AW > Uniform)$ and $PI(RW > Uniform)$ are around 50%, suggesting that our methods match uniform sampling baseline.

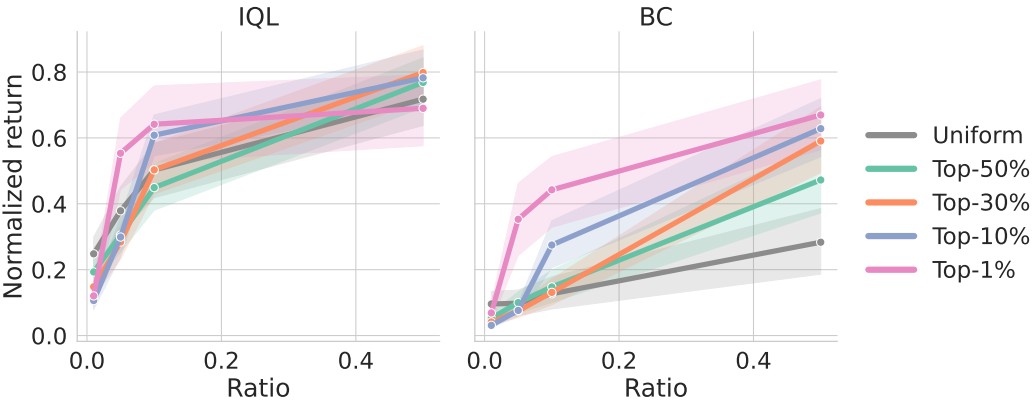

Figure 10: Performance of percentage-filtering sampling with varying percentages.

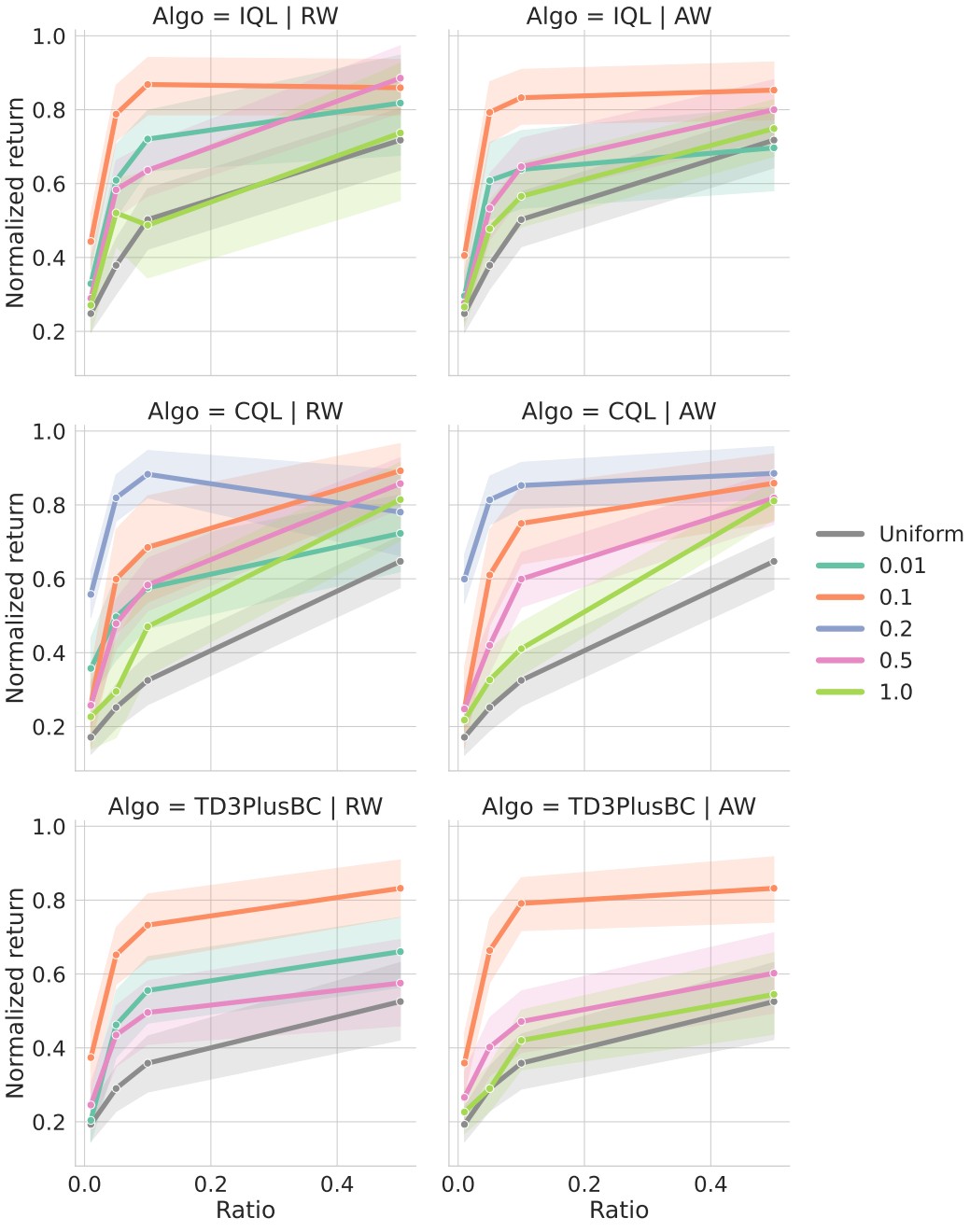

Figure 11: Performance at varying temperature.

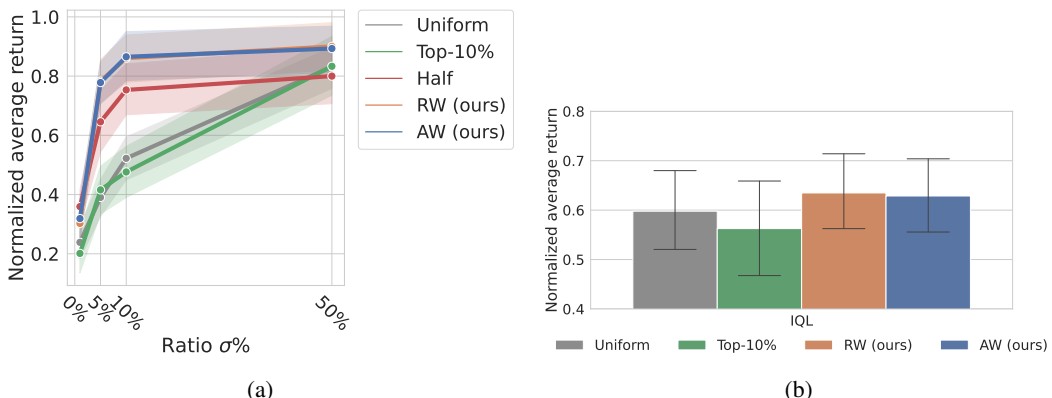

(a)                          (b)

Figure 12: Our results in official implementation shows similar results to the results from our implementation in Sections 5.2 and 5.4. **(a)** The average return of each sampling strategy in mixed dataszets (Section 5.2) in official IQL codebase. We see that our methods also show higher average return than the baselines in similar trend shown in Figure 2, suggesting that the performance gain of our methods are independent of implementation choices. **(b)** The average returns of each sampling methods in regular datasets (Section 5.4). We see that our methods show insignificantly performance in over uniform sampling, similar to the results in Figure 4a.

