# OpenReview forum: "Harnessing Mixed Offline Reinforcement Learning Datasets via Trajectory Weighting"
_ICLR.cc/2023/Conference — ICLR 2023 poster_

### Official Review · Reviewer_Z9su · 2022-10-24

**Confidence:** 4
**Correctness:** 4
**Technical Novelty And Significance:** 3
**Empirical Novelty And Significance:** 3
**Recommendation:** 6

**Clarity, Quality, Novelty And Reproducibility:**

The work is of good quality, overall clear in the explanation. The example code was not provided, but the instructions on environment and realization details are well presented.

**Strength And Weaknesses:**

Strength:

- The novelty is clear. The author is trying to analyze the mixed dataset’s efficiency in a theoretical way, representing and capturing the tendency of PSV.

- The author proposed the two methods in a rigorous logic, return-weighting and advantage weighting are straightforward by the definition.

- The sufficient experiments are evidently showing that the proposed two methods on different baseline methods, could better capture the dataset’s trajectory original feature, and by reweighting the importance of the high-return trajectory, the method’s effectiveness is well improved.

Weakness:

- It’s noticeable from Figure2 that the two weighting methods sometimes have a different appearance, it would be better if the author could analyze the different suitable situations for the above different two methods.

- A little confusing part may be the introduction of negative-sided variance(NSV), because the majority part of the paper is mainly focusing on the RPSV, is NAV the NSV used in the implementation part of two re-weighting methods?


**Summary Of The Paper:**

The paper is trying to solve the problem that when there exists a scenario in the offline RL dataset: Mostly low return trajectory, few high return trajectory, indicating the mixed policies lying behind, the results could be not ideal.

By introducing the RPSV, the variance of the positive case and negative cases are well captured in the original dataset and is helping to infer the distribution of high-return and low-return policies.

And the author base on the variance of PSV proposed two weighting methods. Abundant experiment on CQL, IQL, TD3+BC, sole BC, etc. from various environments like classic control has justified the effectiveness of different methods. Detailed explanations of experiments and deductions are shown in the appendix.


**Summary Of The Review:**

The author is trying to use novel ways to analyze the distribution of offline training datasets. Which helped to improve the performance of the offline RL methods and also justify the applicability to other methods. The paper is presented in an explicit way, also with necessary details from the appendix part.

It’s meaningful work to help offline RL algorithms improve the exploitation in the limited amount of high return trajectories.

---

> ### Author Response · Authors · 2022-11-17
> **response**
>
> We thank the reviewer for acknowledging the novelty and quality of our work and the comprehensiveness of our experiments.
>
> > The example code was not provided
> >
>
> **Answer:**
>
> We already provided the code in the supplementary material (the zip file that is shown on the page). We are sorry for not mentioning this explicitly in the paper. The reproducibility statement in the updated manuscript already indicates that the code is included in the supplementary material.
>
> > It’s noticeable from Figure2 that the two weighting methods sometimes have a different appearance, it would be better if the author could analyze the different suitable situations for the above different two methods.
> >
>
> **Answer:**
>
> There is indeed a visible discrepancy between AW and RW in CQL on the Ant environment, with a mixed dataset with 50% of expert and random data, while on other datasets and algorithms, they have similar performance. It is difficult to draw a concrete conclusion since we only observe a performance discrepancy on one dataset + algorithm pair.
>
> That being said, we want to emphasize the limitation of RW and how AW addresses it. AW aims at handling the randomness at state initialization, as we describe in Section 4.2 and 4.1. If it’s known that, in the environment, every initial state yields similar returns by following a behavioral policy, then there is no advantage of using AW.  Nevertheless, we can construct environments with randomized state initialization that are adversarial to RW, but where the AW approach is sound. For instance, assume that the goal is to navigate without obstacle to reach the upper right of the map, but that the initial state is sampled uniformly on the map and that the behavioral policy is the optimal policy. Then RW will down-weight trajectories that start from initial states that are distant to the goal and up-weight the ones that start close to the goal. We call these trajectories starting close to the goal as trajectories starting from “lucky initial states.” AW intends to correct the risk stemming from over sampling of the lucky initial states.
>
> Nevertheless, as we do not have the ground truth behavioral policy’s expected returns at each initial state, it is difficult to say if the issue is present in D4RL. Indeed, even though D4RL randomizes the initial states, it does not imply that these initial states yield different values when using the same behavioral policy. For instance, if a behavioral policy attains similar returns when starting from all of the initial states, AW will not have benefit over RW since there are no lucky initial states. Overall, we observe that RW yield higher average returns than uniform sampling on most tasks, which seems to imply that the benefits of RW outweigh the risks of RW.
>
> Theoretically, AW can remedy the issue of RW while it could depend on the accuracy of behavioral value estimation. Even though AW might not address the risks of RW, we observe AW performs similarly with RW, which suggests that there is no significant loss of applying AW.
>
> > A little confusing part may be the introduction of negative-sided variance (NSV), because the majority part of the paper is mainly focusing on the RPSV, is NAV the NSV used in the implementation part of two re-weighting methods?
> >
>
> **Answer:**
>
> We were hoping to draw the connection to statistical variance to by showing that NSV + PSV = V. To make it clearer, we have removed NSV from the text in the updated manuscript.

---

### Official Review · Reviewer_BiNt · 2022-10-24

**Confidence:** 4
**Correctness:** 3
**Technical Novelty And Significance:** 2
**Empirical Novelty And Significance:** 2
**Recommendation:** 6

**Clarity, Quality, Novelty And Reproducibility:**

-

**Strength And Weaknesses:**

Strengths:

- The weighted sampling helps combating the need for a near expert dataset for approaches with explicit behavioral cloning terms.

- The weighted sampling approach has been demonstrated to not hurt the performance even for well balance datasets across various modern offline rl approaches (CQL and IQL).


Weakness:

- The results while impressive is not surprising.  While the gains are signficant for BC and TD3+BC, it is well established that these methods will suffer in sparse high reward trajectory settings. The gains over CQL and IQL are marginal for the current D4RL benchmark.

- While the authors introduce RPSV as a measure of tracking the variance in the dataset, it does not answer the question of “When is it necessary to do weighted sampling?”. i.e. The RPSV metric follows a inverted U curve as we increase sigma from 1% to 100% in the artifical datasets that were created.

- For this approach to be applicable, the data needs to be collected such that the trajectories are complete; i.e. The behavioral policies have to start from the starting state distribution. While this is not that hard of a constraint it is to be noted that this approach does not make sense if the collected policies start from different parts of the state-space. (i.e. trajectory stitching settings.)

- Probability of improvement over uniform sampling 3(b) is very skewed compared to the actual amount of improvement we see in figure 3(a). Here it seems that our approach is equally valid for both IQL and TD3+BC according to fig 3(b) whereas 3(a) tells a different story. overall I feel fig 3(b) does not add much value.

Minor points.

- ““However, offline IL requires expert and random data to be separated while we neither assume separated datasets nor access to expert data.” ([pdf](zotero://open-pdf/library/items/BFZJEQLU?page=9)) ” This is not fully accurate as we are using the rewards to rescale sampling we are assuming proxies to both a “separate dataset” and access to “expert data” offline IL does not require a expert data, its just that it wont work without one, as with the case with this approach.-

- sigma has different scale in page 14 and 15. (0.01 and 1 )

- The formatting /Bolding has errors in the tables in appendices.

- It would be wonderful to include plans to release the sampling code snippets for the public.

- IQL numbers for hopper is not matching with the author provided numbers, It might be good idea to doublecheck them.

**Summary Of The Paper:**

This paper explores a particular setting of low sparse high reward trajectories in an offline RL datasets. Offline RL algorithms that explicitly do Behavioral Cloning (BC) have a hard time learning good policies in these settings. The authors propose a simple reweighing scheme to induce an artificial dataset with a higher performing behavioral policy. They also show theoretical analysis regarding the bias of this estimate for deterministic settings and also introduce Return Positive Sided Variance as a measure of quantifying the variance in the returns of trajectories among the dataset. Finally the experimental results show that the weighted sampling of the dataset significantly helps in the artificially generated datasets with varying RPSV however the performance gains on true D4RL tasks are marginal especially for methods without explicit BC like CQL and IQL.

**Summary Of The Review:**

Overall the paper presents a relative simple weighted sampling method which can be a valuable tool in an offline-rl toolkit, however, the metric RPSV presented in the paper fails to capture the scenario where the method can/should be applied. This can be an interesting paper with a better metric / analysis on when the approach should be considered during offline RL, however, under the current version, I am leaning toward a rejection.

---

> ### Author Response · Authors · 2022-11-16
> **response (1/4)**
>
> We thank the reviewer for appreciating the simplicity of our method and acknowledging our work can be a valuable tool in the offline RL toolkit. The reviewer’s questions are addressed in the following.
>
> We first summarize the change we made here:
>
> - We added a section of analysis in Section 5.2 to answer when to use our methods.
> - We changed the x-axis of Figure 2 to help illustrate the best conditions of applying our methods.
>
> The changes are marked in "magenta" in the updated manuscript.
>
> > The results while impressive is not surprising. While the gains are signficant for BC and TD3+BC, it is well established that these methods will suffer in sparse high reward trajectory settings. The gains over CQL and IQL are marginal for the current D4RL benchmark.
> >
>
> **Answer:**
>
> We disagree that the gains over CQL and IQL are marginal in the current D4RL benchmarks. **Note that all of our experiments in original Section 5.2 (now Section 5.3) are carried out in the tasks in D4RL, just with more challenging datasets (mixtures).**
>
> **New Section 5.3 demonstrated our methods’ substantial performance gain when high-return trajectories are sparse (i.e., $\sigma$ is small)**. Each dataset in Section 5.3 is mixed with $\sigma$% expert/medium (high-return) and $1.0-\sigma%$ random (low-return) trajectories. This sort of mixed datasets are likely to be more challenging than regular non-mixed datasets since the majority of trajectories are low-return. We adversarially construct such mixed datasets with $\sigma\%$ from 1%, 5%, 10%, to 50%, to test the capability of our methods in challenging low-return datasets.
>
> **Figure 2 showed that our method has three times better average performance than the baselines** from $\sigma=5%$ to $\sigma=1%$ (RPSV=0.02 in the original submission) and even consistently outperforms the baselines in all $\sigma$ (i.e., expert/medium data ratio). This indicates that our methods exhibit consistent and substantial advantage over the uniform sampling baseline in challenging low-return datasets.
>
> **Furthermore, in Section 5.3, we show that our method doesn’t lose performance** in non-mixed regular datasets. These datasets are less challenging since the average trajectory return in the dataset is higher than mixed datasets we studied. Uniform sampling can already yield satisfactory performance. As our method doesn’t lose performance in such “easier” datasets, one can always apply it irrespective of the datasets.
>
> **Finally, we would like to stress that the current D4RL benchmark cannot test algorithms’ performance in datasets with only a handful of high-return trajectories.** We believe this sort of datasets is important as more natural and should be of interest to the offline RL community since collecting high-return trajectories is tedious while gathering random trajectories is easy.

---

> ### Author Response · Authors · 2022-11-16
> **response (2/4)**
>
> > While the authors introduce RPSV as a measure of tracking the variance in the dataset, it does not answer the question of “When is it necessary to do weighted sampling?”. i.e. The RPSV metric follows a inverted U curve as we increase sigma from 1% to 100% in the artifical datasets that were created.
> >
>
> **Answer:**
>
> When **high-performing trajectories are sparse in datasets**, using our weighted sampling methods is preferable/necessary. Thus, we performed experiments in Section 5.3 with mixed datasets consisting of random trajectories (low-return) and varying ratios $\sigma$ of expert/medium trajectories (high-return).
>
> **Figure 2 shows our methods achieve higher performance than baselines at low expert/medium data ratios.** We changed the x-axis of Figure 2 to better illustrate our methods’ advantages over the baselines (e.g., Uniform, Top-10%, etc.). New Figure 2 shows that our methods (AW and RW) exhibit considerable performance gain over uniform sampling (gray curve) and also outperform other baselines, given low ratios of expert/medium data ($\sigma$) ($1\% \leq \sigma\% \leq 10\%$). Remarkably, AW and RW can exceed the performance of an algorithm trained with complete expert datasets (denoted as dashed lines). As our methods’ performances plateau at $\sigma \approx 10\%$, we only tested if our methods retain their good performance at higher ratios of expert/medium data ($\sigma =50\%$), showing that our methods maintain their advantage when the number of high-return trajectories increases. Note that “Half” is a new baseline added during rebuttal, and requires extra assumption compared to our method. “Half” assumes the knowledge of source (i.e., from expert/medium or from random) of each trajectory in the dataset, and samples 50% data from expert/medium and random datasets, respectively, in each batch of gradient updates. The details can be found in Section 5.1.
>
> **We hypothesize that our methods’ performance gain over uniform sampling results from increased RPSV in the datasets.** The design of a robust predictor for the performance gain is not trivial since offline RL’s performance are influenced by several factors, including the environment and the offline RL algorithm it’s paired with. Here we focus on two statistical factors that are easy to estimate from the dataset: (i) the mean return of a dataset and (ii) RPSV. Although dependent from each other, these two factors have a good variability in our experiments since increasing the ratio of expert/medium data would not only increase RPSV but also the mean return of a dataset.
>
> We show the relationship between the performance gain over uniform sampling of an algorithm (represented by the color of the dot in the plots below), datasets’ mean return (x-axis), and RPSV (y-axis, in log scale) in Figure 3. Each dot denotes the average performance gain in a tuple of environment, dataset, and $\sigma$.  It can be seen that at similar mean returns (x-axis), our methods’ performance gain grow evidently (color gets closer to red) when RPSV increases (y-axis). The performance gain at high RPSV can be related to the performance gain with low $\sigma$ (expert/medium data ratio) in Figure 2 since most datasets with low mean returns have high RPSV.
>
> Based on the above figures and analysis, we conclude that the advantage of our method is the highest when RPSV is high. We also notice that a high dataset mean return may temper its advantage. The reason is that uniform sampling baseline is already quite efficient in the settings where $\sigma%$ is in a high range such as 50%, and that the room for additional improvement over it is therefore limited. This is not the result of a failure of our resampling strategy but of a success of the uniform sampling one.
>
> As a summary,  **it is preferable to use our weighted sampling method when a dataset has low mean return and high RPSV** since it is likely to attain better performance than uniform sampling.
>
>
> > For this approach to be applicable, the data needs to be collected such that the trajectories are complete; i.e. The behavioral policies have to start from the starting state distribution. While this is not that hard of a constraint it is to be noted that this approach does not make sense if the collected policies start from different parts of the state-space. (i.e. trajectory stitching settings.)
> >
>
> **Answer:**
>
> We acknowledge this is an assumption of our method. Though this assumption is, as the reviewer said, not that hard of a constraint, we have discussed it and its possible solutions/workarounds in the new manuscript.

---

> ### Author Response · Authors · 2022-11-16
> **response (3/4)**
>
> > Probability of improvement over uniform sampling 3(b) is very skewed compared to the actual amount of improvement we see in figure 3(a). Here it seems that our approach is equally valid for both IQL and TD3+BC according to fig 3(b) whereas 3(a) tells a different story. overall I feel fig 3(b) does not add much value.
> >
>
> **Answer:**
>
> We would like to first clarify that original Figure 3a (now Figure 4a) is the result of regular (non-mixed) datasets while Figure 3b (now Figure 7) shows the results in both non-mixed and mixed datasets. **Hence, the shape in the original Figure 3a cannot be directly translated to the original Figure 3b.**
>
> **To avoid confusion, we plot in Figure 7 the probability of improvements (PI) separately in regular (upper row) and mixed datasets (lower row).** It can be seen that in the Figure 7, the lower bounds of confidence intervals (the error bar) in our methods are clearly higher than 0.5, suggesting statistically significant improvements over uniform sampling (according to [1]) in mixed datasets with sparse high-return trajectories. Even in regular datasets, the mean PIs of our methods are all above 50%. This indicates that our methods are not worse than uniform sampling on average in regular datasets that have more high-return trajectories than mixed datasets. Hence, a user can **expect our methods yield performance higher than or equal to uniform sampling baseline irrespective to datasets.** Our method attains the highest gain over baseline when only a few high-return trajectories are in the dataset, while matching the baseline’s performance when plenty of high-return trajectories are in the dataset.
>
> **We want to highlight that probability of improvements (PI) measures the robustness of a method, conveying different messages than average performance shown in Figure 2 and Figure 4a.** In Figure 7, PI measures “how likely is a method to perform better than uniform sampling in a randomly selected environment, dataset, and random seed?” PI captures the uncertainty among random seeds while aggregated metrics like average performance doesn’t. For example, suppose we have 5 trials with different random seeds on the same environment and dataset for two methods A and B. The fact that A has higher average return than B, doesn’t follow that A always performs better than B in all trials. It is possible that A is worse than B in some trials. Comparing only the average return, one would mis-conclude that A is certainly better than B. Instead, PI answers “how likely is A to be better than B?”
>
> **PI is important for algorithm selection since it measures the robustness of a method.** One can have extremely high performance gain in a few tasks and lose to baselines in the majority of tasks. If so, this new method would not always outperform baselines, which makes it not robust. A robust method should consistently improve baselines and not lose performance in most tasks.
>
> **Robustness of a method is important for a user to decide whether or not to prefer the new method over the existing method (i.e., baselines).** As offline RL algorithms’ performance interplay with several factors (e.g., dataset properties, environment dynamics, reward functions, etc), it is unlikely to accurately predict what conditions make the new method perform the best. Lacking of perfect knowledge of the best condition for the new method, it is unclear whether a user should deploy the new method on a new task that doesn’t have benchmarking results yet. As a result, robustness is crucial when selecting an algorithm for a new task. If the new method is shown to be robust and perform better than the baseline in most trials (i.e., high PI), it would be worth preferring the new method over the baseline. In contrast, it’s not worth using the new method if the new method has PI below 50%.
>
> [1] Agarwal, Rishabh, et al. "Deep reinforcement learning at the edge of the statistical precipice." *Advances in neural information processing systems*
>  34 (2021): 29304-29320.

---

> ### Author Response · Authors · 2022-11-16
> **response (4/4)**
>
> > ““However, offline IL requires expert and random data to be separated while we neither assume separated datasets nor access to expert data.” (pdf) ” This is not fully accurate as we are using the rewards to rescale sampling we are assuming proxies to both a “separate dataset” and access to “expert data” offline IL does not require a expert data, its just that it wont work without one, as with the case with this approach.
> >
>
> **Answer:**
>
> Thanks for pointing this out. We didn’t intend to claim that our methods require “fewer” assumptions than offline IL, but rather “different” assumptions. Our definition about offline IL is based on the recent works (Kim et al., 2021; Ma et al., 2022; Xu et al., 2022, see reference). We clarify the reviewer’s questions in the following.
>
> > we are using the rewards to rescale sampling we are assuming proxies to both a “separate dataset” and access to “expert data”
> >
>
> It should be noted that access to rewards doesn’t imply that the dataset has to be separated. We say a dataset is separated if we can divide a dataset into to two subsets: expert and non-expert. According to Xu et al., 2022, offline IL methods require the dataset to be separated since they train a discriminator to classify if a trajectory is from experts. However, our method doesn’t require a dataset to be separated and hence is applicable when we don’t know the sources of each trajectory. For example, the replay dataset in d4rl is collected by policies at different time points during training, and cannot be separated into expert and non-expert subsets. As such offline IL is not trivially applicable in replay datasets, while our methods are.
>
> > offline IL does not require a expert data, its just that it wont work without one, as with the case with this approach.
> >
>
> Our methods don’t require expert data to outperform uniform sampling baseline. We summarize the experimental result in mixed medium datasets in the following table. It can be seen that AW and RW exhibit higher performance over uniform sampling, suggesting that expert data is not a requirement for our method to improve over the baseline.
>
> |         | BC       | CQL      | IQL      | TD3+BC    |
> |:-------:|----------|----------|----------|-----------|
> | Uniform | 0.14     | 0.43     | 0.52     | 0.34      |
> |    AW   | **0.50** | **0.67** | **0.58** | **0.622** |
> |    RW   | **0.50** | **0.67** | **0.67** | **0.60**  |
>
>
> That being said, we agree that our wording is confusing. To be effective, both offline IL and our methods require some advantageous data in a suboptimal dataset consisting of low-performing (i.e., low-return or non-expert) data. It’s just that these advantageous data points don’t have to be expert data (they just need to be better than the low-performing ones).
>
> We rephrased this paragraph in the updated manuscript to make the comparison clearer and more precise.
>
> > sigma has different scale in page 14 and 15. (0.01 and 1 )
> >
>
> **Answer:**
>
> We correct it in the updated manuscript! Thanks for pointing this out.
>
> > The formatting /Bolding has errors in the tables in appendices.
> >
>
> **Answer:**
>
> We are not sure which part of bolding/formatting is wrong. Could the reviewer clarify it? Here we explain again our formatting scheme (detailed in Appendix A.9). In each row of Tables 2, 3, 4, and 5 in appendices, we bold a number if it is higher than the performance of uniform sampling, and add an asterisk symbol when the number is the highest in the row.
>
> > It would be wonderful to include plans to release the sampling code snippets for the public.
> >
>
> **Answer:**
>
> Our original submission already included the source code in the supplementary material. It will be made public on acceptance. We’re sorry for not mentioning code release in the paper. We have mentioned this in the updated manuscript.
>
> > IQL numbers for hopper is not matching with the author provided numbers, It might be good idea to doublecheck them.
> >
>
> **Answer:**
>
> Thanks for pointing out this. We’ve compared our implementation (based on d3rlpy) and the official implementation of IQL. The major difference is the base deep learning library: ours is based on pytorch and the official IQL is based on jax. To ensure the results is independent of IQL implementation, we’ve launched new runs of our experiments using the official codebase and will update the result during rebuttal.

---

> > ### Author Response · Authors · 2022-11-19
> > **Experiments finished**
> >
> > We want to update our response to the following question:
> > > IQL numbers for hopper is not matching with the author provided numbers, It might be good idea to doublecheck them.
> >
> > We have already finished the experiments in the official IQL codebase and presented the results in Section. A13. The results show the similar trend with the results from our implementation.

---

> > ### Comment · Reviewer_BiNt · 2022-11-28
> > **Response to Rebuttal.**
> >
> > The authors have been very throrough in addressing the issues pointed out in the original review. All the clarifying comments and figures are most welcome. I have updated my score.
> >
> > ps: While Figure 3 makes it much clearer on "when the proposed approach improves performance", it still begs the question if we could answer this question if the average performance was calculated from returns normalized by maximum return seen in the dataset rather than oracle provided information. This way, it will be a more clearer "measure" of when uniform sampling may fail.

---

> > > ### Author Response · Authors · 2022-12-09
> > > **Response**
> > >
> > > We thank the reviewer for raising the score. We would like to further clarify a comment from the reviewer in the following.
> > >
> > > > ps: While Figure 3 makes it much clearer on "when the proposed approach improves performance", it still begs the question if we could answer this question if the average performance was calculated from returns normalized by maximum return seen in the dataset rather than oracle provided information. This way, it will be a more clearer "measure" of when uniform sampling may fail.
> > > >
> > >
> > > **Answer:** Instead of normalizing by the maximum return seen in the dataset, the returns in Figure 3 (color of dots) are normalized by the average returns of expert and random policies (see Section 5.1). We believe this normalization scheme uses the *“oracle provided information”* to normalize returns. This normalization scheme is also used in the paper that proposes the D4RL benchmark ([https://arxiv.org/pdf/2004.07219.pdf](https://arxiv.org/pdf/2004.07219.pdf)).

---

### Official Review · Reviewer_C84m · 2022-10-25

**Confidence:** 4
**Correctness:** 3
**Technical Novelty And Significance:** 2
**Empirical Novelty And Significance:** 2
**Recommendation:** 6

**Clarity, Quality, Novelty And Reproducibility:**

The idea proposed in the paper is novel, as far as I know. But the paper lacks comparison to simple baseline that is commonly used to tackle this problem. The caption of the figure does not clearly explain the reasoning as to why the information shown demonstrates that the proposed method outperforms baselines.

**Strength And Weaknesses:**

Strength

The proposed idea is intuitive and the claim that the idea is useful when there are many more low return episodes than high return episodes makes sense.

Weaknesses

Lack of comparison to simple baseline. The issue that the paper studies is well-known, and a common technique to tackle it is to ensure each batch of episodes used to update the networks contain half successful, and half failed episodes. I would have liked to see the comparison to this simple and common baselines.

Parts of the writing are not clear. For example:

- "cold start performance boost" in the Introduction.

- "Most offline RL algorithms are anchored to the behavior policy". What are anchors mean, and where is the evidence that support this claims?

- How should I interpret figure 2? The caption states that the proposed methods outperform baselines, but how is this conclusion supported by the figure?

**Summary Of The Paper:**

The paper proposes to reweight episodes in the replay by their advantage values. The paper claims that doing so is especially useful when there are many more low return episodes in the replay buffer compared to high return episodes.

**Summary Of The Review:**

I look forward to the rebuttal. The idea is simple, and may therefore be used widely. But as it currently stands, it is unclear if the method outperforms simple and common baselines.

---

> ### Author Response · Authors · 2022-11-16
> **response (1/2)**
>
> We thank the reviewer for acknowledging the simplicity, novelty, and potential wide applicability of our work. We compare our methods with the simple baselines suggested by the reviewer and clarify the writing issues in the following.
>
> > Lack of comparison to simple baseline
> >
>
> **Answer:**
>
> In addition to top-10% which was part of the submission, we include now the baseline that samples half from high-return (i.e., expert or medium) and low-return (i.e., random) datasets in each batch of network update in Figure 2 and illustrate its implementation in Section 5.1. Note that we also changed the x-axis of Figure 2 to $\sigma$ (i.e., the ratio of expert/medium trajectories in a mixed dataset).
>
> The new baseline is termed as “Half” in Figure 2. We see that “Half” attains a higher average return (y-axis) than uniform sampling, while “Half” loses to our methods (AW and RW) at all ratio $\sigma\%$ (x-axis). This suggests that our method is more effective than “Half” in mixed datasets. Note that “Half” is not tested in pure random dataset (i.e., $\sigma\%=0$)  and non-mixed regular datasets since “Half” requires a dataset to consist a subset of high-performing and a subset of low-performing trajectories, which are flagged as such. This requirement further makes the strength our method more evident since our methods don’t require the knowledge of the dataset separation.
>
> > *What are anchors mean*
> >
>
> **Answer:**
>
> The concept of anchor in offline RL is novel to this paper. An anchor is what allows a boat to remain in place at sea around some location. In this analogy, we claim that most offline RL algorithms would work as follows: they drop an anchor at the (estimated) behavioral policy location and then propel the boat in order to maximize their main objective. Dropping this anchor makes sure that optimization errors (such as strong currents at sea) cannot impair too much the performance of the trained policy as compared to that of the behavioral policy.
>
> The fact that most offline RL algorithms use a form of anchoring is folklore knowledge. It is supported by the distribution shift paradox [1]: in order to improve the behavioral policy, one must change it, but by changing it, we shift the induced state-action occupancy measure, which means that the evaluation of the trained policy becomes an out-of-distribution problem. As a result, most algorithms resort to some form of *anchoring regularization*: penalizing policies with high divergence with the behavioral policy (Laroche et al., 2019; Fujimoto et al., 2019; Fujimoto & Gu, 2021; Kumar et al., 2019a), preventing to take rare actions (Fujimoto et al. 2019), penalizing actions that were rarely taken in the current states (pessimism) (Petrik et al., 2016; Kumar et al., 2020b; Buckman et al., 2020), etc.
>
> Further, Section 5.2 in the recent theoretical work ([https://arxiv.org/pdf/2009.06799.pdf](https://arxiv.org/pdf/2009.06799.pdf)) shows that in most of the deep offline RL methods, target policies are constrained to be close to the behavioral policy, and hence the performance of the target policies are likely to be close to the mean return of the dataset.
>
> Finally, our empirical evidence in Figure 6 in the Appendix corroborates this statement, showing that offline RL performance drops when the performance of the behavioral policy decreases.
>
> [1] Levine, Sergey, et al. "Offline reinforcement learning: Tutorial, review, and perspectives on open problems." *arXiv preprint arXiv:2005.01643*
>  (2020).
>
> > *Cold start performance*
> >
>
> **Answer:**
>
> Cold start performance usually refers to the performance of a trained model at its initialization. Again, the behavioral policy is where the anchor is dropped and somehow the starting point of the offline RL algorithms. As a consequence, when we allow ourselves to select a behavioral policy with a better performance, we change its cold start performance. More specifically, since the weights assigned to the trajectories implicitly define a behavioral policy, the mean return of a dataset implicitly defines the performance of a behavioral policy (see Section 4.1 for details).
>
> Quote from our original paper, *“by giving larger weights to high-return trajectories, we can control the implicit behavior policy to be high performing and therefore grant a cold start performance boost to the offline RL algorithm.”* Prior works [[https://arxiv.org/pdf/2009.06799.pdf](https://arxiv.org/pdf/2009.06799.pdf)] and our work (see Figure 6 in Appendix) show that offline RL algorithms return policies that correlate to the performance of the behavioral policy. Re-weighting trajectories using our method increases the performance of the behavioral policy implicitly defined by the weights of trajectories, hence providing a better cold start performance (i.e., the lower bound of offline RL performance).

---

> > ### Comment · Reviewer_C84m · 2022-11-19
> > **Response to rebuttal**
> >
> > I have increased my score to 6 due to the addition of the “Half" experiment.
> >
> > Regarding language, I still do not believe the authors should invent their own terminology and metaphor to explain the behavior of current algorithms, especially when these behaviors are well known (such as the learnt policy being constrained to the behavior policy).

---

> ### Author Response · Authors · 2022-11-16
> **response (2/2)**
>
> > *How should I interpret figure 2? The caption states that the proposed methods outperform baselines, but how is this conclusion supported by the figure?*
> >
>
> **Answer:**
>
> In Figure 2, we aim to show that our methods can achieve higher average return than the baselines. The x-axis is the ratio of expert/medium data (denoted as $\sigma \%$); the y-axis is the average normalized return obtained by training an offline RL algorithm (i.e., CQL, IQL, BC, and TD3+BC) with a sampling strategy (i.e., Uniform, Top-10%, Half, RW, and AW). Figure 2 shows that our methods are able to achieve near expert (near 1.0) performance with low ratios of expert/medium data in the mixed dataset, and consistently yield higher return than all baselines. Thus, our methods outperform the baselines with both low and high ratios of expert/medium data.

---

### Official Review · Reviewer_56VB · 2022-10-29

**Confidence:** 4
**Correctness:** 4
**Technical Novelty And Significance:** 3
**Empirical Novelty And Significance:** 4
**Recommendation:** 8

**Clarity, Quality, Novelty And Reproducibility:**

About Novelty:
- This problem itself is not novel, as a lot of work in the past has been about solving the distribution skew in offline datasets. But their main approaches are by skewing the sampling methods. Thus the idea of introducing a weight in sampled data is novel.

About Organization:
- I feel that the paper’s structure is very clear and concise, from preliminary to results. From what the problem is, and why the problem needs to be addressed, all the way to the author’s methods.
- Section 3 has a lot of reasons for the motivation of the work, especially in figure 1, the analysis is clear and very insightful.

**Strength And Weaknesses:**

Main concern:
- How prevalent is this problem in offline RL? I see the table in A.3, but it does not give the whole image of this problem. Is this problem seen in other mainstream datasets?
- How does this method compare to the other sampling strategy methods? Why should this method be used over the other ones?
- Would making the model learn the policy on offline datasets make the model overfit?
- Is this method robust under different distributions of dataset? For example in Figure 1, what if the dataset has two density spikes, on 0.0, and 0.8.
- In the result part, there is no comparison between the two methods, the authors only gave experiments on their method vs the vanilla method of no weighted distributions.

**Summary Of The Paper:**

The authors tackle a long-known problem in offline learning where the performance of the RL agent is highly dependent on the distribution of high and low return trajectories. It follows that in datasets that contain a high percentage of low-returning trajectories and a low percentage of high-returning trajectories. The Agent will be overly restrained by the low-return trajectories.

The authors tackle this problem by implementing trajectory weighting, where different weights are assigned to different trajectories of different importance, naturally a more important trajectory would have a more important weight. After assigning weights to each trajectory, the authors add an entropy regularization term.

**Summary Of The Review:**

About Challenge:
- Without collecting additional data, how can the performance of the policy be improved? Previous works have tackled this by skewing their sampling.

About Contribution:
- The authors reweight the transitions in the dataset, and regularize it by adding an entropy regularization term. Then they further change the reweighting term to equation 11 in the paper.

---

> ### Author Response · Authors · 2022-11-17
> **response (1/2)**
>
> We thank the reviewer for appreciating our paper and acknowledging our contribution. We answer the rest of the questions in the following.
>
> > How prevalent is this problem in offline RL? I see the table in A.3, but it does not give the whole image of this problem. Is this problem seen in other mainstream datasets?
> >
>
> **Answer:**
>
> We believe that real world scenario often involve situations where a lot of low quality data has been collected alongside to a little high quality data. It may also happen that the data quality (novice, intermediate, expert) is logged, for instance in online gaming where players have a public ranking. As a consequence, we claim that enabling offline RL to efficiently learn from such heterogenous datasets with limited amount of high-return trajectories is important for deploying offline RL in more realistic tasks.
>
> Regarding the usefulness of the method to other mainstream datasets, this is not something we have looked into. There could be a long off-topic discussion about these datasets and how little diverse they tend to be, repeating local randomizations of the same trajectories in comparison with what would be encountered in real life with data collected from many real humans. In any case, we found our methods to be slightly (insignificantly) advantageous even in these “regular” dataset.
>
>
> > How does this method compare to the other sampling strategy methods? Why should this method be used over the other ones?
> >
>
> > In the result part, there is no comparison between the two methods, the authors only gave experiments on their method vs the vanilla method of no weighted distributions.
> >
>
> **Answer:**
>
> In addition to uniform sampling, we also compared our methods with **two other sampling strategies in Sections 5.2 and 5.4:**
>
> 1. One baseline is **“Top-k%”** sampling strategy that samples transitions only from top k% trajectories sorted by returns. In Figure 2, we present the results of Top-10% since it works the best across all methods.
> 2. During rebuttal, we added a new baseline **“Half”** that samples half of transitions from expert/medium subset and half transitions from random subset of a mixed dataset. A mixed dataset consists of $\sigma\%$ of expert/medium trajectories and ($1-\sigma$%) random trajectories.
>
> The updated Section 5.2 show that our methods achieve higher average return in all datasets with varying $\sigma$, suggesting that our methods have better performance on average. Moreover, our methods are not extremely sensitive to temperature setting. In Figure 8 in Appendix, we see that our methods are able to improve over uniform sampling with a wide range of temperatures. Hence, given a dataset with low mean return and high RPSV, one should use our method over the other baselines.
>
> In order to avoid confusion, we would like to also point out that other sample weighting strategies found in the RL literature address completely different purposes such as model error minimization [Schaul2015], online policy optimization [Laroche2021], off-policy policy evaluation [Precup2000,Thomas2015], etc., and these methods would not be applicable to our setting.
>
> [Precup2000] Precup, Doina. "Eligibility traces for off-policy policy evaluation." *Computer Science Department Faculty Publication Series* (2000): 80.
>
> [Schaul2015] Schaul, Tom, et al. "Prioritized experience replay." *arXiv preprint arXiv:1511.05952* (2015).
>
> [Thomas2015] Thomas, Philip S. "Safe reinforcement learning." (2015).
>
> [Laroche2021] Laroche, Romain, and Remi Tachet des Combes. "Dr Jekyll & Mr Hyde: the strange case of off-policy policy updates." *Advances in Neural Information Processing Systems* 34 (2021): 24442-24454.
>
> > Would making the model learn the policy on offline datasets make the model overfit?
> >
>
> **Answer:**
>
> We did not observe obvious overfitting caused by our method in the experiments. As our datasets of interest consist of only handful of high-return trajectories, if the learned target policy does not generalize, it is unlikely to demonstrate good performance (i.e., near 1.0 in normalized return). Our experiments show that in our methods achieve higher average return than the uniform sampling baseline. This indicates that deploying our method in offline RL does not magnify the risk of overfitting.
>
> That being said, it is possible that training a target policy using offline RL in some datasets might have unavoidable overfitting. For example, training a target policy in a pure random dataset is likely to result in overfitting to state-action coverage of the random policy and hence lead to poor performance since the policy never experiences states from high-return trajectories. Yet, we want to emphasize that overfitting in such datasets does not result from our methods or sampling strategy but from the nature of the dataset.

---

> ### Author Response · Authors · 2022-11-17
> **response (2/2)**
>
> > Is this method robust under different distributions of dataset? For example in Figure 1, what if the dataset has two density spikes, on 0.0, and 0.8.
> >
>
> **Answer:**
>
> We plot the return distribution of all datasets in Figure 7 in Appendix, showing that mixed datasets with 50% expert/medium data ($\sigma=50$) have two density pikes. In such datasets, our methods are still able to achieve average returns higher than or equal to uniform sampling baseline. This indicates that our method doesn’t suffer in datasets with two density spikes on returns.
>
> > This problem itself is not novel, as a lot of work in the past has been about solving the distribution skew in offline datasets. But their main approaches are by skewing the sampling methods. Thus the idea of introducing a weight in sampled data is novel.
> >
>
> **Answer:**
>
> Could the reviewer point out some references? To our best knowledge, the problem of learning from datasets with skew distribution of returns was not studied in offline RL literature.
>
> Our guess is that the reviewer is referring to off-policy evaluation methods with importance sampling (IS) [1,2,3,4]. These methods re-weight the optimization objective of policy/value function using importance weighting to correct off-policy errors. The importance weighting captures how close the current policy is to the samples generated by other policies (i.e., behavioral policies). This line of works are completely different both in functioning and in purpose.
>
> In functioning, IS intends to re-weight the rewards based on the similarity to the importance weighting to the behavioral policy (i.e., the ratio of target policy to the behavioral policy), while AW/RW re-weight the trajectory distribution based on the trajectory returns.
>
> In purposes, IS is purposed for correcting off-policy errors in the training samples, while AW/RW is designed for making a better behavioral policy for offline RL algorithm to stick with. Offline RL algorithms, as we discussed in Section 3.2, often need to regularize the target policy to be close to the behavioral policy that collects the dataset. AW/RW changes the behavioral policy by re-weighting trajectories in the dataset.
>
> [1] Hanna, Josiah, Scott Niekum, and Peter Stone. "Importance sampling policy evaluation with an estimated behavior policy." *International Conference on Machine Learning*. PMLR, 2019.
>
> [2] Xie, Yuan, et al. "Off-policy evaluation and learning from logged bandit feedback: Error reduction via surrogate policy." International Conference on Representation Learning (ICLR), 2019.
>
> [3] Thomas, Philip, and Emma Brunskill. "Data-efficient off-policy policy evaluation for reinforcement learning." *International Conference on Machine Learning,* PMLR, 2016.
>
> [4] Yang, Mengjiao, et al. "Off-policy evaluation via the regularized lagrangian." *Advances in Neural Information Processing Systems,* 33 (2020): 6551-6561.

---

### Author Response · Authors · 2022-11-19
**General response**

We thank the reviewers for reviewing our paper and would like kindly please the reviewer to take a look at our response in the individual review. Also, we summarize our response in the following (the changes in the paper are marked in magenta).

- **Code release:** We already included the **source code in the supplementary material** of our **original submission** and indicated the code access in the reproducibility statement in the updated manuscript.
- **New baseline:** We added a new baseline suggested by Reviewer *C84m.* This baseline samples half of the transitions from expert/medium and random datasets. In the updated Figure 2, we show that **our methods achieve higher average return than the baseline at all expert/medium data ratios**.
- **New analysis:** We added a new analysis in Section 5.3 to verify if our methods benefit from high RPSV in datasets. **This new analysis shows that our methods’ performance gain over uniform sampling baseline grows when RPSV increases**.
- **Prevalence/Importance of mixed datasets:**  We believe that real-world scenarios often involve situations where **a lot of low-quality data has been collected alongside a little high-quality data**. It may also happen that the data quality (novice, intermediate, expert) is logged, for instance, in online gaming where players have a public ranking. As a consequence, we claim that enabling offline RL to efficiently learn from such heterogenous datasets with a limited amount of high-return trajectories is important for deploying offline RL in more realistic tasks.
- **Results in official IQL codebase:** We additionally ran the experiments in Sections 5.2 and 5.3 with the official IQL codebase, showing in Figure 12 in the appendix, **similar amounts of a performance gain as we showed in our IQL codebase**.

---

### Decision · Program_Chairs · 2023-01-20

**Decision:**

Accept: poster

**Justification For Why Not Higher Score:**

The novelty is limited, the results are not surprising, and the lack of comparison with certain baselines.

**Justification For Why Not Lower Score:**

Although the novelty of the method is limited, overall, it is a well-executed paper that could be useful for the community.

**Metareview: Summary, Strengths And Weaknesses:**

The paper studies an important problem, has a novel and intuitive idea (although its novelty is rather limited), is well-written, and the method is supported by the right amount of experimental results (although the results are not that surprising). Overall, it is a well-executed paper that could be useful for the community. I would thank the authors for responding to the reviewers' comments and made changes in their paper. I would suggest they take the rest of the reviewers' comments into consideration in preparing the final version of the paper.


**Note From Pc:**

if the above contains the word "oral" or "spotlight" please see: "oral" presentation means -> notable-top-5% and "spotlight" means -> notable-top-25%. As stated in our emails, we are disassociating presentation type from AC recommendations